# Towards Stable Test-time Adaptation in Dynamic Wild World

**Shuaicheng Niu**[14]*, **Jiaxiang Wu**[2]*, **Yifan Zhang**[3]*, **Zhiquan Wen**[1], **Yaofo Chen**[1],
**Peilin Zhao**[2], **Mingkui Tan**[15]†
sensc@mail.scut.edu.cn; mingkuitan@scut.edu.cn
South China University of Technology[1]   Tencent AI Lab[2]   National University of Singapore[3]
Key Laboratory of Big Data and Intelligent Robot, Ministry of Education[4]   Pazhou Laboratory[5]

## Abstract

Test-time adaptation (TTA) has shown to be effective at tackling distribution shifts between training and testing data by adapting a given model on test samples. However, the online model updating of TTA may be unstable and this is often a key obstacle preventing existing TTA methods from being deployed in the real world. Specifically, TTA may fail to improve or even harm the model performance when test data have: 1) mixed distribution shifts, 2) small batch sizes, and 3) online imbalanced label distribution shifts, which are quite common in practice. In this paper, we investigate the unstable reasons and find that the batch norm layer is a crucial factor hindering TTA stability. Conversely, TTA can perform more stably with batch-agnostic norm layers, *i.e.*, group or layer norm. However, we observe that TTA with group and layer norms does not always succeed and still suffers many failure cases. By digging into the failure cases, we find that certain noisy test samples with large gradients may disturb the model adaption and result in collapsed trivial solutions, *i.e.*, assigning the same class label for all samples. To address the above collapse issue, we propose a sharpness-aware and reliable entropy minimization method, called SAR, for further stabilizing TTA from two aspects: 1) remove partial noisy samples with large gradients, 2) encourage model weights to go to a flat minimum so that the model is robust to the remaining noisy samples. Promising results demonstrate that SAR performs more stably over prior methods and is computationally efficient under the above wild test scenarios. The source code is available at https://github.com/mr-eggplant/SAR.

## 1 Introduction

Deep neural networks achieve excellent performance when training and testing domains follow the same distribution (He et al., 2016; Wang et al., 2018; Choi et al., 2018). However, when domain shifts exist, deep networks often struggle to generalize. Such domain shifts usually occur in real applications, since test data may unavoidably encounter natural variations or corruptions (Hendrycks & Dietterich, 2019; Koh et al., 2021), such as the weather changes (*e.g.*, *snow, frost, fog*), sensor degradation (*e.g.*, *Gaussian noise, defocus blur*), and many other reasons. Unfortunately, deep models can be sensitive to the above shifts and suffer from severe performance degradation even if the shift is mild (Recht et al., 2018). However, deploying a deep model on test domains with distribution shifts is still an urgent demand, and model adaptation is needed in these cases.

Recently, numerous test-time adaptation (TTA) methods (Sun et al., 2020; Wang et al., 2021; Iwasawa & Matsuo, 2021; Bartler et al., 2022) have been proposed to conquer the above domain shifts by online updating a model on the test data, which include two main categories, *i.e.*, Test-Time Training (TTT) (Sun et al., 2020; Liu et al., 2021) and Fully TTA (Wang et al., 2021; Niu et al., 2022a). In this work, we focus on Fully TTA since it is more generally to be used than TTT in two aspects: i) it does not alter training and can adapt arbitrary pre-trained models to the test data without access to original training data; ii) it may rely on fewer backward passes (only one or less than one) for each test sample than TTT (see efficiency comparisons of TTT, Tent and EATA in Table 6).

---

*Equal contribution. †Corresponding author. Work done when S. Niu works as an intern in Tencent AI Lab.

Figure 1: An illustration of practical/wild test-time adaptation (TTA) scenarios, in which prior online TTA methods may degrade severely. The accuracy of Tent (Wang et al., 2021) is measured on ImageNet-C of level 5 with ResNet50-BN (15 mixed corruptions in *(a)* and Gaussian in *(b-c)*).

TTA has been shown boost model robustness to domain shifts significantly. However, its excellent performance is often obtained under some mild test settings, *e.g.*, adapting with a batch of test samples that have the same distribution shift type and randomly shuffled label distribution (see Figure 1 ①). In the complex real world, test data may come arbitrarily. As shown in Figure 1 ②, the test scenario may meet: i) mixture of multiple distribution shifts, ii) small test batch sizes (even single sample), iii) the ground-truth test label distribution $Q_t(y)$ is online shifted and $Q_t(y)$ may be imbalanced at each time-step $t$. In these wild test settings, online updating a model by existing TTA methods may be unstable, *i.e.*, failing to help or even harming the model's robustness.

To stabilize wild TTA, one immediate solution is to recover the model weights after each time adaptation of a sample or mini-batch, such as MEMO (Zhang et al., 2022) and episodic Tent (Wang et al., 2021). Meanwhile, DDA (Gao et al., 2022) provides a potentially effective idea to address this issue: rather than model adaptation, it seeks to transfer test samples to the source training distribution (via a trained diffusion model (Dhariwal & Nichol, 2021)), in which all model weights are frozen during testing. However, these methods cannot cumulatively exploit the knowledge of previous test samples to boost adaptation performance, and thus obtain limited results when there are lots of test samples. In addition, the diffusion model in DDA is expected to have good generalization ability and can project any possible target shifts to the source data. Nevertheless, this is hard to be satisfied as far as it goes, *e.g.*, DDA performs well on *noise* shifts while less competitive on *blur* and *weather* (see Table 2). Thus, how to stabilize online TTA under wild test settings is still an open question.

In this paper, we first point out that the batch norm (BN) layer (Ioffe & Szegedy, 2015) is a key obstacle since under the above wild scenarios the mean and variance estimation in BN layers will be biased. In light of this, we further investigate the effects of norm layers in TTA (see Section 4) and find that pre-trained models with batch-agnostic norm layers (*i.e.*, group norm (GN) (Wu & He, 2018) and layer norm (LN) (Ba et al., 2016)) are more beneficial for stable TTA. However, TTA on GN/LN models does not always succeed and still has many failure cases. Specifically, GN/LN models optimized by online entropy minimization (Wang et al., 2021) tend to occur collapse, *i.e.*, predicting all samples to a single class (see Figure 2), especially when the distribution shift is severe. To address this issue, we propose a **s**harpness-**a**ware and **r**eliable entropy minimization method (namely SAR). Specifically, we find that indeed some noisy samples that produce gradients with large norms harm the adaptation and thus result in model collapse. To avoid this, we filter partial samples with large and noisy gradients out of adaptation according to their entropy. For the remaining samples, we introduce a sharpness-aware learning scheme to ensure that the model weights are optimized to a flat minimum, thereby being robust to the large and noisy gradients/updates.

**Main Findings and Contributions.** (1) We analyze and empirically verify that batch-agnostic norm layers (*i.e.*, GN and LN) are more beneficial than BN to stable test-time adaptation under wild test settings, *i.e.*, mix domain shifts, small test batch sizes and online imbalanced label distribution shifts (see Figure 1). (2) We further address the model collapse issue of test-time entropy minimization on GN/LN models by proposing a sharpness-aware and reliable (SAR) optimization scheme, which jointly minimizes the entropy and the sharpness of entropy of those reliable test samples. SAR is simple yet effective and enables online test-time adaptation stabilized under wild test settings.

## 2  PRELIMINARIES

We revisit two main categories of test-time adaptation methods in this section for the convenience of further analyses, and put detailed related work discussions into Appendix A due to page limits.

**Test-time Training (TTT).** Let $f_\Theta(\mathbf{x})$ denote a model trained on $\mathcal{D}_{train} = \{(\mathbf{x}_i, y_i)\}_{i=1}^N$ with parameter $\Theta$, where $\mathbf{x}_i \in \mathcal{X}_{train}$ (the training data space) and $y_i \in \mathcal{C}$ (the label space). The goal of test-time adaptation (Sun et al., 2020; Wang et al., 2021) is to boost $f_\Theta(\mathbf{x})$ on out-of-distribution test samples $\mathcal{D}_{test} = \{\mathbf{x}_j\}_{j=1}^M$, where $\mathbf{x}_j \in \mathcal{X}_{test}$ (testing data space) and $\mathcal{X}_{test} \neq \mathcal{X}_{train}$. Sun et al. (2020) first propose the TTT pipeline, in which at **training phase** a model is trained on source $\mathcal{D}_{train}$ via both cross-entropy $\mathcal{L}_{CE}$ and self-supervised rotation prediction (Gidaris et al., 2018) $\mathcal{L}_S$:

$$\min_{\Theta_b, \Theta_c, \Theta_s} \mathbb{E}_{\mathbf{x} \in \mathcal{D}_{train}} [\mathcal{L}_{CE}(\mathbf{x}; \Theta_b, \Theta_c) + \mathcal{L}_S(\mathbf{x}; \Theta_b, \Theta_s)], \tag{1}$$

where $\Theta_b$ is the task-shared parameters (shadow layers), $\Theta_c$ and $\Theta_s$ are task-specific parameters (deep layers) for $\mathcal{L}_{CE}$ and $\mathcal{L}_S$, respectively. At **testing phase**, given a test sample $\mathbf{x}$, TTT first updates the model with self-supervised task: $\Theta'_b \leftarrow \arg\min_{\Theta_b} \mathcal{L}_S(\mathbf{x}; \Theta_b, \Theta_s)$ and then use the updated model weights $\Theta'_b$ to perform final prediction via $f(\mathbf{x}; \Theta'_b, \Theta_c)$.

**Fully Test-time Adaptation (TTA).** The pipeline of TTT needs to alter the original model training process, which may be infeasible when training data are unavailable due to privacy/storage concerns. To avoid this, Wang et al. (2021) propose fully TTA, which adapts arbitrary pre-trained models for a given test mini-batch by conducting entropy minimization (**Tent**): $\min - \sum_c \hat{y}_c \log \hat{y}_c$ where $\hat{y}_c = f_\Theta(c|\mathbf{x})$ and $c$ denotes class $c$. This method is more efficient than TTT as shown in Table 6.

## 3 STABLE ADAPTATION BY TEST ENTROPY AND SHARPNESS MINIMIZATION

**Test-time Adaptation (TTA) in Dynamic Wild World.** Although prior TTA methods have exhibited great potential for out-of-distribution generalization, its success may rely on some underlying test prerequisites (as illustrated in Figure 1): 1) test samples have the same distribution shift type; 2) adapting with a batch of samples each time, 3) the test label distribution is uniform during the whole online adaptation process, which, however, are easy to be violated in the wild world. In wild scenarios (Figure 1 ②), prior methods may perform poorly or even fail. In this section, we seek to analyze the underlying reasons why TTA fails under wild testing scenarios described in Figure 1 from a unified perspective (c.f. Section 3.1) and then propose associated solutions (c.f. Section 3.2).

### 3.1 WHAT CAUSES UNSTABLE TEST-TIME ADAPTATION?

We first analyze why wild TTA fails by investigating the norm layer effects in TTA and then dig into the unstable reasons for entropy-based methods with batch-agnostic norms, *e.g.*, group norm.

**Batch Normalization Hinders Stable TTA.** In TTA, prior methods often conduct adaptation on pre-trained models with batch normalization (BN) layers (Ioffe & Szegedy, 2015), and most of them are built upon BN statistics adaptation (Schneider et al., 2020; Nado et al., 2020; Khurana et al., 2021; Wang et al., 2021; Niu et al., 2022a; Hu et al., 2021; Zhang et al., 2022). Specifically, for a layer with $d$-dimensional input $\mathbf{x} = \left(x^{(1)} \dots x^{(d)}\right)$, the batch normalized output are: $y^{(k)} = \gamma^{(k)}\hat{x}^{(k)} + \beta^{(k)}$, where $\hat{x}^{(k)} = \left(x^{(k)} - \mathrm{E}\left[x^{(k)}\right]\right)/\sqrt{\mathrm{Var}\left[x^{(k)}\right]}$. Here, $\gamma^{(k)}$ and $\beta^{(k)}$ are learnable affine parameters. BN adaptation methods calculate mean $\mathrm{E}[x^{(k)}]$ and variance $\mathrm{Var}[x^{(k)}]$ over (a batch of) **test samples.** However, in wild TTA, all three practical adaptation settings (in Figure 1) in which TTA may fail will result in problematic mean and variance estimation. **First**, BN statistics indeed represent a distribution and ideally each distribution should have its own statistics. Simply estimating shared BN statistics of multiple distributions from mini-batch test samples unavoidably obtains limited performance, such as in multi-task/domain learning (Wu & Johnson, 2021). **Second**, the quality of estimated statistics relies on the batch size, and it is hard to use very few samples (*i.e.*, small batch size) to estimate it accurately. **Third**, the imbalanced label shift will also result in biased BN statistics towards some specific classes in the dataset. Based on the above, we posit that batch-agnostic norm layers, *i.e.*, agnostic to the way samples are grouped into a batch, are more suitable for performing TTA, such as group norm (GN) (Wu & He, 2018) and layer norm (LN) (Ba et al., 2016). We devise our method based on GN/LN models in Section 3.2.

To verify the above claim, we empirically investigate the effects of different normalization layers (including BN, GN, and LN) in TTA (including TTT and Tent) in Section 4. From the results, we observe that models equipped with GN and LN are more stable than models with BN when

Figure 2: Failure case analyses (a-c) of online test-time entropy minimization (Wang et al., 2021). (a) and (b) record the model predictions during online adaptation. (c) illustrates how gradients norm evolves with and without model collapse. (d) investigates the relationship between the sample's entropy and gradients norm. All experiments are conducted on shuffled ImageNet-C of Gaussian noise with ResNet50 (GN), and a larger (severity) level denotes a more severe distribution shift.

performing online test-time adaptation under three practical test settings (in Figure 1) and have fewer failure cases. The detailed empirical studies are put in Section 4 for the coherence of presentation.

**Online Entropy Minimization Tends to Result in Collapsed Trivial Solutions, *i.e.*, Predict All Samples to the Same Class.** Although TTA performs more stable on GN and LN models, it does not always succeed and still faces several failure cases (as shown in Section 4). For example, entropy minimization (Tent) on GN models (ResNet50-GN) tends to collapse, especially when the distribution shift extent is severe. In this paper, we aim to stabilize online fully TTA under various practical test settings. To this end, we first analyze the failure reasons, in which we find models are often optimized to collapse trivial solutions. We illustrate this issue in the following.

During the online adaptation process, we record the predicted class and the gradients norm (produced by entropy loss) of ResNet50-GN on shuffled ImageNet-C of Gaussian noise. By comparing Figures 2 (a) and (b), entropy minimization is shown to be unstable and may occur collapse when the distribution shift is severe (*i.e.*, severity level 5). From Figure 2 (a), as the adaptation goes by, the model tends to predict all input samples to the same class, even though these samples have different ground-truth classes, called model collapse. Meanwhile, we notice that along with the model starts to collapse the $\ell_2$-norm of gradients of all trainable parameters suddenly increases and then degrades to almost 0 (as shown in Figure 2 (c)), while on severity level 3 the model works well and the gradients norm keep in a stable range all the time. This indicates that some test samples produce large gradients that may hurt the adaptation and lead to model collapse.

### 3.2 SHARPNESS-AWARE AND RELIABLE TEST-TIME ENTROPY MINIMIZATION

Based on the above analyses, two most straightforward solutions to avoid model collapse are filtering out test samples according to the sample gradients or performing gradients clipping. However, these are not very feasible since the gradients norms for different models and distribution shift types have different scales, and thus it is hard to devise a general method to set the threshold for sample filtering or gradient clipping (see Section 5.2 for more analyses). We propose our solutions as follows.

**Reliable Entropy Minimization.** Since directly filtering samples with gradients norm is infeasible, we first investigate the relation between entropy loss and gradients norm and seek to remove samples with large gradients based on their entropy. Here, the entropy depends on the model's output class number $C$ and it belongs to $(0, \ln C)$ for different models and data. In this sense, the threshold for filtering samples with entropy is easier to select. As shown in Figure 2 (d), selecting samples with small loss values can remove part of samples that have large gradients (area@1) out of adaptation. Formally, let $E(\mathbf{x}; \Theta)$ be the entropy of sample $\mathbf{x}$, the selective entropy minimization is defined by:

$$\min_{\Theta} S(\mathbf{x})E(\mathbf{x}; \Theta), \quad \text{where} \quad S(\mathbf{x}) \triangleq \mathbb{I}_{\{E(\mathbf{x};\Theta)<E_0\}}(\mathbf{x}). \tag{2}$$

Here, $\Theta$ denote model parameters, $\mathbb{I}_{\{\cdot\}}(\cdot)$ is an indicator function and $E_0$ is a pre-defined parameter. Note that the above criteria will also remove samples within area@2 in Figure 2 (d), in which the samples have low confidence and thus are unreliable (Niu et al., 2022a).

**Sharpness-aware Entropy Minimization.** Through Eqn. (2), we have removed test samples in area@1&2 in Figure 2 (d) from adaptation. Ideally, we expect to optimize the model via samples only in area@3, since samples in area@4 still have large gradients and may harm the adaptation. However, it is hard to further remove the samples in area@4 via a filtering scheme. Alternatively,

Figure 3: Batch size effects of different TTA methods under different models (different normalization layers). Experiments are conducted on ImageNet-C of Gaussian noise. We report mean and standard deviation of 3 runs with different random seeds. 'na' denotes no adapt accuracy. Note that except for Vit-LN, the standard deviation is too small to display in the figures.

we seek to make the model insensitive to the large gradients contributed by samples in `area@4`. Here, we encourage the model to go to a flat area of the entropy loss surface. The reason is that a flat minimum has good generalization ability and is robust to noisy/large gradients, *i.e.*, the noisy/large updates over the flat minimum would not significantly affect the original model loss, while a sharp minimum would. To this end, we jointly minimize the entropy and the sharpness of entropy loss by:

$$\min_{\Theta} E^{SA}(\mathbf{x};\Theta), \quad \text{where} \quad E^{SA}(\mathbf{x};\Theta) \triangleq \max_{\|\boldsymbol{\epsilon}\|_2 \leq \rho} E(\mathbf{x};\Theta+\boldsymbol{\epsilon}). \tag{3}$$

Here, the inner optimization seeks to find a weight perturbation $\boldsymbol{\epsilon}$ in a Euclidean ball with radius $\rho$ that maximizes the entropy. The sharpness is quantified by the maximal change of entropy between $\Theta$ and $\Theta + \boldsymbol{\epsilon}$. This bi-level problem encourages the optimization to find flat minima. To address problem (3), we follow SAM (Foret et al., 2021) that first approximately solves the inner optimization via first-order Taylor expansion, *i.e.*,

$$\boldsymbol{\epsilon}^*(\Theta) \triangleq \arg\max_{\|\boldsymbol{\epsilon}\|_2 \leq \rho} E(\mathbf{x};\Theta+\boldsymbol{\epsilon}) \approx \arg\max_{\|\boldsymbol{\epsilon}\|_2 \leq \rho} E(\mathbf{x};\Theta)+\boldsymbol{\epsilon}^T \nabla_{\Theta} E(\mathbf{x};\Theta) = \arg\max_{\|\boldsymbol{\epsilon}\|_2 \leq \rho} \boldsymbol{\epsilon}^T \nabla_{\Theta} E(\mathbf{x};\Theta).$$

Then, $\hat{\boldsymbol{\epsilon}}(\Theta)$ that solves this approximation is given by the solution to a classical dual norm problem:

$$\hat{\boldsymbol{\epsilon}}(\Theta) = \rho\, \text{sign}\left(\nabla_{\Theta} E(\mathbf{x};\Theta)\right) |\nabla_{\Theta} E(\mathbf{x};\Theta)| / \left\|\nabla_{\Theta} E(\mathbf{x};\Theta)\right\|_2. \tag{4}$$

Substituting $\hat{\boldsymbol{\epsilon}}(\Theta)$ back into Eqn. (3) and differentiating, by omitting the second-order terms for computation acceleration, the final gradient approximation is:

$$\nabla_{\Theta} E^{SA}(\mathbf{x};\Theta) \approx \nabla_{\Theta} E(\mathbf{x};\Theta)\big|_{\Theta+\hat{\boldsymbol{\epsilon}}(\Theta)}. \tag{5}$$

**Overall Optimization.** In summary, our sharpness-aware and reliable entropy minimization is:

$$\min_{\tilde{\Theta}} S(\mathbf{x}) E^{SA}(\mathbf{x};\Theta), \tag{6}$$

where $S(\mathbf{x})$ and $E^{SA}(\mathbf{x};\Theta)$ are defined in Eqns. (2) and (3) respectively, $\tilde{\Theta} \subset \Theta$ denote learnable parameters during test-time adaptation. In addition, to avoid a few extremely hard cases that Eqn. (6) may also fail, we further introduce a **Model Recovery Scheme**. We record a moving average $e_m$ of entropy loss values and reset $\tilde{\Theta}$ to be original once $e_m$ is smaller than a small threshold $e_0$, since models after occurring collapse will produce very small entropy loss. Here, the additional memory costs are negligible since we only optimize affine parameters in norm layers (see Appendix C.2 for more details). We summarize the details of our method in Algorithm 1 in Appendix B.

# 4 EMPIRICAL STUDIES OF NORMALIZATION LAYER EFFECTS IN TTA

This section designs experiments to illustrate how test-time adaptation (TTA) performs on models with different norm layers (including BN, GN and LN) under wild test settings described in Figure 1. We verify two representative methods introduced in Section 2, *i.e.*, self-supervised **TTT** (Sun et al., 2020) and unsupervised **Tent** (a fully TTA method) (Wang et al., 2021). Considering that the norm layers are often coupled with mainstream network architectures, we conduct adaptation on ResNet-50-BN (R-50-BN), ResNet-50-GN (R-50-GN) and VitBase-LN (Vit-LN). All adopted model weights are public available and obtained from `torchvision` or `timm` repository (Wightman, 2019). Implementation details of experiments in this section can be found in Appendix C.2.

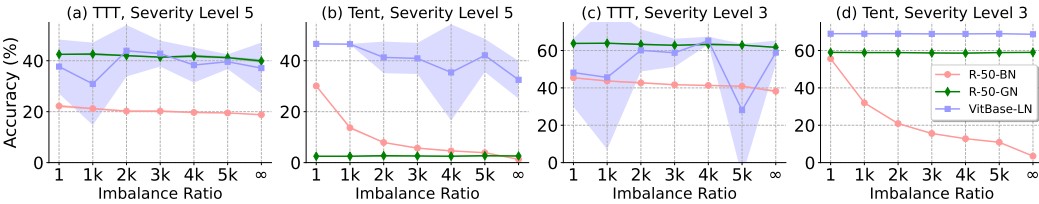

Figure 4: Performance of TTA methods on different models (different norm layers) under the mixture of 15 different corruption types (ImageNet-C). We report mean&stdev. over 3 independent runs.

Figure 5: Performance of TTA methods with different models (different norm layers) under online imbalanced label distribution shifts on ImageNet-C (Gaussian noise). We report mean&stdev. results of 3 runs. Note that except for VitBase-LN, the stdev. is too small to display in the figures.

**(1) Norm Layer Effects in TTA Under Small Test Batch Sizes.** We evaluate TTA methods (TTT and Tent) with different batch sizes (BS), selected from $\{1, 2, 4, 8, 16, 32, 64\}$. Due to GPU memory limits, we only report results of BS up to 8 or 16 for TTT (in original TTT BS is 1), since TTT needs to augment each test sample multiple times (for which we set to 20 by following Niu et al. (2022a)).

From Figure 3, we have: **i)** For Tent, compared with R-50-BN, R-50-GN and Vit-LN are less sensitive to small test batch sizes. The adaptation performance of R-50-BN degrades severely when the batch size goes small ($<8$), while R-50-GN/Vit-LN show stable performance across various batch sizes (Vit-LN on levels 5&3 and R-50-GN on level 3, in subfigures (b)&(d)). It is worth noting that Tent with R-50-GN and Vit-LN not always succeeds and also has failure cases, such as R-50-GN on level 5 (Tent performs worse than no adapt), which is analyzed in Section 3.1. **ii)** For TTT, all R-50-BN/GN and Vit-LN can perform well under various batch sizes. However, TTT with Vit-LN is very unstable and has a large variance over different runs, showing that TTT+VitBase is very sensitive to different sample orders. Here, TTT performs well with R-50-BN under batch size 1 is mainly benefited from TTT applying multiple data augmentations to a single sample to form a mini-batch.

**(2) Norm Layer Effects in TTA Under Mixed Distribution Shifts.** We evaluate TTA methods on models with different norm layers when test data come from multiple shifted domains simultaneously. We compare 'no adapt', 'avg. adapt' (the average accuracy of adapting on each domain separately) and 'mix adapt' (adapting on mixed and shifted domains) accuracy on ImageNet-C consisting of 15 corruption types. The larger accuracy gap between 'mix adapt' and 'avg. adapt' indicates the more sensitive to mixed distribution shifts.

From Figure 4, we have: **i)** For both Tent and TTT, R-50-GN and Vit-LN perform more stable than R-50-BN under mix domain shifts. Specifically, the mix adapt accuracy of R-50-BN is consistently poor than the average adapt accuracy across different severity levels (in all subfigures (a-d)). In contrast, R-50-GN and Vit-LN are able to achieve comparable accuracy of mix and average adapt, *i.e.*, TTT on R-50-GN (levels 5&3) and Tent on Vit-LN (level 3). **ii)** For R-50-GN and Vit-LN, TTT performs more stable than Tent. To be specific, Tent gets 3/4 failure cases (R-50-GN on levels 5&3, Vit-LN on level 5), which is more than that of TTT. **iii)** The same as Section 4 (1), TTT on Vit-LN has large variances over multiple runs, showing TTT+Vit-LN is sensitive to different sample orders.

**(3) Norm Layer Effects in TTA Under Online Imbalanced Label Shifts.** As in Figure 1 (c), during the online adaptation process, the label distribution $Q_t(y)$ at different time-steps $t$ may be different (online shift) and imbalanced. To evaluate this, we first simulate this imbalanced label distribution shift by adjusting the order of input samples (from a test set) as follows.

*Online Imbalanced Label Distribution Shift Simulation.* Assuming that we have totally $T$ time-steps and $T$ equals to the class number $C$. We set the probability vector $Q_t(y) = [q_1, q_2, ..., q_C]$, where $q_c = q_{max}$ if $c = t$ and $q_c = q_{min} \triangleq (1 - q_{max})/(C - 1)$ if $c \neq t$. Here, $q_{max}/q_{min}$ denotes the imbalance ratio. Then, at each $t \in \{1, 2, ..., T = C\}$, we sample $M$ images from the test set according to $Q_t(y)$. Based on ImageNet-C (Gaussian noise), we construct a new testing set that has

online imbalanced label distribution shifts with totally $100(M) \times 1000(T)$ images. Note that we pre-shuffle the class orders in ImageNet-C, since we cannot know which class will come in practice.

From Figure 5, we have: **i)** For Tent, R-50-GN and Vit-LN are less sensitive than R-50-BN to online imbalanced label distribution shifts (see subfigures (b)&(d)). Specifically, the adaptation accuracy of R-50-BN (levels 5&3) degrades severely as the imbalance ratio increases. In contrast, R-50-GN and Vit-LN have the potential to perform stably under various imbalance ratios (*e.g.*, R-50-GN and Vit-LN on level 3). **ii)** For TTT, all R-50-BN/GN and Vit-LN perform relatively stable under label shifts, except for TTT+Vit-LN has large variances. The adaptation accuracy will also degrade but not very severe as the imbalance ratio increases. **iii)** Tent with GN is more sensitive to the extent of distribution shift than BN. Specifically, for imbalanced ratio 1 (all $Q_t(y)$ are uniform) and severity level 5, Tent+R-50-GN fails and performs poorer than no adapt, while Tent+R-50-BN works well.

**(4) Overall Observations.** Based on all the above results, we have: **i)** R-50-GN and Vit-LN are more stable than R-50-BN when performing TTA under wild test settings (see Figure 1). However, they do not always succeed and still suffer from several failure cases. **ii)** R-50-GN is more suitable for self-supervised TTT than Vit-LN, since TTT+Vit-LN is sensitive to different sample orders and has large variances over different runs. **iii)** Vit-LN is more suitable for unsupervised Tent than R-50-GN, since Tent+R-50-GN is easily to collapse, especially when the distribution shift is severe.

## 5 COMPARISON WITH STATE-OF-THE-ARTS

**Dataset and Methods.** We conduct experiments based on ImageNet-C (Hendrycks & Dietterich, 2019), a large-scale and widely used benchmark for out-of-distribution generalization. It contains 15 types of 4 main categories (noise, blur, weather, digital) corrupted images and each type has 5 severity levels. We compare our SAR with the following state-of-the-art methods. DDA (Gao et al., 2022) performs input adaptation at test time via a diffusion model. MEMO (Zhang et al., 2022) minimizes marginal entropy over different augmented copies w.r.t. a given test sample. Tent (Wang et al., 2021) and EATA (Niu et al., 2022a) are two entropy based online fully test-time adaptation (TTA) methods.

Table 1: Efficiency comparison for processing 50,000 images (Gaussian noise, level 5 on ImageNet-C) via a single V100 GPU on ResNet50-GN.

| Method | GPU time |
|---|---|
| MEMO (Zhang et al., 2022) | 55,980 secs |
| DDA (Gao et al., 2022) | 146,220 secs |
| TTT (Sun et al., 2020) | 3,600 secs |
| Tent (Wang et al., 2021) | 110 secs |
| EATA (Niu et al., 2022a) | 114 secs |
| SAR (ours) | 115 secs |

**Models and Implementation Details.** We conduct experiments on ResNet50-BN/GN and VitBase-LN that are obtained from `torchvision` or `timm` (Wightman, 2019). For our SAR, we use SGD as the update rule, with a momentum of 0.9, batch size of 64 (except for the experiments of batch size=1), and learning rate of 0.00025/0.001 for ResNet/Vit models. The threshold $E_0$ in Eqn. (2) is set to $0.4 \times \ln 1000$ by following EATA (Niu et al., 2022a). $\rho$ in Eqn. (3) is set by the default value 0.05 in Foret et al. (2021). For trainable parameters of SAR during TTA, following Tent (Wang et al., 2021), we **adapt the affine parameters** of group/layer normalization layers in ResNet50-GN/VitBase-LN. More details and hyper-parameters of compared methods are put in Appendix C.2.

### 5.1 ROBUSTNESS TO CORRUPTION UNDER VARIOUS WILD TEST SETTINGS

**Results under Online Imbalanced Label Distribution Shifts.** As illustrated in Section 4, as the imbalance ratio $q_{max}/q_{min}$ increases, TTA degrades more and more severe. Here, we make comparisons under the most difficult case: $q_{max}/q_{min}=\infty$, *i.e.*, test samples come in class order. We evaluate all methods under different corruptions via the same sample sequence for fair comparisons.

From Table 2, our SAR achieves the best results in average of 15 corruption types over ResNet50-GN and VitBase-LN, suggesting its effectiveness. It is worth noting that Tent works well for many corruption types on VitBase-LN (*e.g.*, *defocus* and *motion blur*) and ResNet50-GN (*e.g.*, *pixel*), while consistently fails on ResNet50-BN. This further verifies our observations in Section 4 that entropy minimization on LN/GN models has the potential to perform well under online imbalanced label distribution shifts. Meanwhile, Tent also suffers from many failure cases, *e.g.*, VitBase-LN on *shot noise* and *snow*. For these cases, our SAR works well. Moreover, EATA has fewer failure cases than Tent and achieves higher average accuracy, which indicates that the weight regularization is somehow able to alleviate the model collapse issue. Nonetheless, the performance of EATA is still inferior to our SAR, *e.g.*, 49.9% *vs.* 58.0% (ours) on VitBase-LN regarding average accuracy.

Table 2: Comparisons with state-of-the-art methods on ImageNet-C (severity level 5) under **ONLINE IMBALANCED LABEL SHIFTS** (imbalance ratio = ∞) regarding **Accuracy (%)**. "BN"/"GN"/"LN" is short for Batch/Group/Layer normalization. The **bold** number indicates the best result.

| Model+Method | Noise | | | Blur | | | | Weather | | | | Digital | | | | Avg. |
|---|---|---|---|---|---|---|---|---|---|---|---|---|---|---|---|---|
| | Gauss. | Shot | Impul. | Defoc. | Glass | Motion | Zoom | Snow | Frost | Fog | Brit. | Contr. | Elastic | Pixel | JPEG | |
| ResNet50 (BN) | 2.2 | 2.9 | 1.8 | 17.8 | 9.8 | 14.5 | 22.5 | 16.8 | 23.4 | 24.6 | 59.0 | 5.5 | 17.1 | 20.7 | 31.6 | 18.0 |
| • MEMO | 7.4 | 8.6 | 8.9 | 19.8 | 13.2 | 20.8 | 27.5 | 25.6 | 28.6 | 32.3 | 60.8 | 11.0 | 23.8 | 33.2 | 37.7 | 24.0 |
| • DDA | 32.2 | 33.1 | 32.0 | 14.6 | 16.4 | 16.6 | 24.4 | 20.0 | 25.5 | 17.2 | 52.2 | 3.2 | 35.7 | 41.8 | 45.4 | 27.2 |
| • Tent | 1.2 | 1.4 | 1.4 | 1.0 | 0.9 | 1.2 | 2.6 | 1.7 | 1.8 | 3.6 | 5.0 | 0.5 | 2.6 | 3.2 | 3.1 | 2.1 |
| • EATA | 0.3 | 0.3 | 0.3 | 0.2 | 0.2 | 0.5 | 0.9 | 0.8 | 0.9 | 1.8 | 3.5 | 0.2 | 0.8 | 1.2 | 0.9 | 0.9 |
| ResNet50 (GN) | 17.9 | 19.9 | 17.9 | **19.7** | 11.3 | 21.3 | 24.9 | 40.4 | **47.4** | 33.6 | 69.2 | 36.3 | 18.7 | 28.4 | 52.2 | 30.6 |
| • MEMO | 18.4 | 20.6 | 18.4 | 17.1 | 12.7 | 21.8 | 26.9 | **40.7** | 46.9 | 34.8 | 69.6 | 36.4 | 19.2 | 32.2 | 53.4 | 31.3 |
| • DDA | **42.5** | **43.4** | **42.3** | 16.5 | 19.4 | 21.9 | 26.1 | 35.8 | 40.2 | 13.7 | 61.3 | 25.2 | **37.3** | 46.9 | 54.3 | 35.1 |
| • Tent | 2.6 | 3.3 | 2.7 | 13.9 | 7.9 | 19.5 | 17.0 | 16.5 | 21.9 | 1.8 | 70.5 | 42.2 | 6.6 | 49.4 | 53.7 | 22.0 |
| • EATA | 27.0 | 28.3 | 28.1 | 14.9 | 17.1 | 24.4 | 25.3 | 32.2 | 32.0 | 39.8 | 66.7 | 33.6 | 24.5 | 41.9 | 38.4 | 31.6 |
| • SAR (ours) | 33.1±1.0 | 36.5±0.4 | 35.5±1.1 | 19.2±0.4 | **19.5±1.2** | **33.3±0.5** | **27.7±4.0** | 23.9±5.1 | 45.3±0.4 | **50.1±1.0** | **71.9±0.1** | **46.7±0.2** | 7.1±1.8 | **52.1±0.5** | **56.3±0.1** | **37.2±0.6** |
| VitBase (LN) | 9.4 | 6.7 | 8.3 | 29.1 | 23.4 | 34.0 | 27.0 | 15.8 | 26.3 | 47.4 | 54.7 | 43.9 | 30.5 | 44.5 | 47.6 | 29.9 |
| • MEMO | 21.6 | 17.4 | 20.6 | 37.1 | 29.6 | 40.6 | 34.4 | 25.0 | 34.8 | 55.2 | 65.0 | 54.9 | 37.4 | 55.5 | 57.7 | 39.1 |
| • DDA | 41.3 | 41.3 | 40.6 | 24.6 | 27.4 | 30.7 | 26.9 | 18.2 | 27.7 | 34.8 | 50.0 | 32.3 | 42.2 | 52.5 | 52.7 | 36.2 |
| • Tent | 32.7 | 1.4 | 34.6 | 54.4 | 52.3 | 58.2 | 52.2 | 7.7 | 12.0 | 69.3 | 76.1 | 66.1 | 56.7 | 69.4 | 66.4 | 47.3 |
| • EATA | 35.9 | 34.6 | 36.7 | 45.3 | 47.2 | 49.3 | 47.7 | **56.5** | **55.4** | 62.2 | 72.2 | 21.7 | 56.2 | 64.7 | 63.7 | 49.9 |
| • SAR (ours) | **46.5±3.0** | **43.1±7.4** | **48.9±0.4** | **55.3±0.1** | **54.3±0.2** | **58.9±0.1** | **54.8±0.2** | 53.6±7.1 | 46.2±3.5 | **69.7±0.1** | **76.2±0.1** | **66.2±0.3** | **60.9±0.3** | **69.6±0.1** | **66.6±0.1** | **58.0±0.5** |

Table 4: Comparisons with state-of-the-art methods on ImageNet-C (severity level 5) with **BATCH SIZE=1** regarding **Accuracy (%)**. "BN"/"GN"/"LN" is short for Batch/Group/Layer normalization.

| Model+Method | Noise | | | Blur | | | | Weather | | | | Digital | | | | Avg. |
|---|---|---|---|---|---|---|---|---|---|---|---|---|---|---|---|---|
| | Gauss. | Shot | Impul. | Defoc. | Glass | Motion | Zoom | Snow | Frost | Fog | Brit. | Contr. | Elastic | Pixel | JPEG | |
| ResNet50 (BN) | 2.2 | 2.9 | 1.9 | 17.9 | 9.8 | 14.8 | 22.5 | 16.9 | 23.3 | 24.4 | 58.9 | 5.4 | 17.0 | 20.6 | 31.6 | 18.0 |
| • MEMO | 7.5 | 8.7 | 8.9 | 19.7 | 13.0 | 20.8 | 27.6 | 25.4 | 28.7 | 32.2 | 60.9 | 11.0 | 23.8 | 32.9 | 37.5 | 23.9 |
| • DDA | 32.1 | 32.8 | 31.8 | 14.7 | 16.6 | 16.6 | 24.2 | 20.0 | 25.4 | 17.2 | 52.1 | 3.2 | 35.7 | 41.5 | 45.3 | 27.3 |
| • Tent | 0.1 | 0.1 | 0.1 | 0.1 | 0.1 | 0.1 | 0.2 | 0.2 | 0.2 | 0.2 | 0.2 | 0.1 | 0.1 | 0.2 | 0.1 | 0.1 |
| • EATA | 0.1 | 0.1 | 0.1 | 0.1 | 0.1 | 0.1 | 0.2 | 0.2 | 0.2 | 0.1 | 0.2 | 0.1 | 0.1 | 0.2 | 0.1 | 0.1 |
| ResNet50 (GN) | 18.0 | 19.8 | 17.9 | **19.8** | 11.4 | 21.4 | 24.9 | 40.4 | **47.3** | 33.6 | 69.3 | 36.3 | 18.6 | 28.4 | 52.3 | 30.6 |
| • MEMO | 18.5 | 20.5 | 18.4 | 17.1 | 12.6 | 21.8 | 26.9 | 40.4 | 47.0 | 34.4 | 69.5 | 36.5 | 19.2 | 32.1 | 53.3 | 31.2 |
| • DDA | **42.4** | **43.3** | **42.3** | 16.6 | **19.6** | 21.9 | 26.0 | 35.7 | 40.1 | 13.7 | 61.2 | 25.2 | **37.5** | 46.6 | 54.1 | 35.1 |
| • Tent | 2.5 | 2.9 | 2.5 | 13.5 | 3.6 | 18.6 | 17.6 | 15.3 | 23.0 | 1.4 | 70.4 | 42.2 | 6.2 | **49.2** | 53.8 | 21.5 |
| • EATA | 24.8 | 28.3 | 25.7 | 18.1 | 17.3 | 28.5 | 29.3 | 44.5 | 44.3 |  | 70.9 | 44.6 | 27.0 | 46.8 | 55.7 | **36.5** |
| • SAR (ours) | 23.4±0.3 | 26.6±0.4 | 23.9±0.0 | 18.4±0.1 | 15.4±0.3 | 28.6±0.3 | 30.4±0.2 | **44.9±0.4** | 44.7±0.2 | 25.7±0.6 | **72.3±0.2** | 44.5±0.1 | 14.8±2.7 | 47.0±0.1 | **56.1±0.2** | 34.5±0.2 |
| VitBase (LN) | 9.5 | 6.7 | 8.2 | 29.0 | 23.4 | 33.9 | 27.1 | 15.9 | 26.5 | 47.2 | 54.7 | 44.1 | 30.5 | 44.5 | 47.8 | 29.9 |
| • MEMO | 21.6 | 17.3 | 20.6 | 37.1 | 29.6 | 40.4 | 34.4 | 24.9 | 34.7 | 55.1 | 64.8 | 54.9 | 37.4 | 55.5 | 57.7 | 39.1 |
| • DDA | 41.3 | **41.1** | 40.7 | 24.4 | 27.2 | 30.6 | 26.9 | 18.3 | 27.5 | 34.6 | 50.1 | 32.4 | 42.3 | 52.2 | 52.6 | 36.1 |
| • Tent | **42.2** | 1.0 | **43.3** | 52.4 | 48.2 | 55.5 | 50.5 | 16.5 | 16.9 | 66.4 | **74.9** | 64.7 | 51.6 | 67.0 | 64.3 | 47.7 |
| • EATA | 29.7 | 25.1 | 34.6 | 44.7 | 39.2 | 48.3 | 42.4 | 37.5 | 45.9 | 60.0 | 65.9 | 61.2 | 46.4 | 58.2 | 59.6 | 46.6 |
| • SAR (ours) | 40.8±0.4 | 36.4±0.7 | 41.5±0.3 | **53.7±0.2** | **50.7±0.1** | **57.5±0.1** | **52.8±0.3** | **59.1±0.4** | **50.7±0.6** | **68.1±1.1** | 74.6±0.7 | **65.7±0.6** | **57.9±0.1** | **68.9±0.1** | **65.9±0.0** | **56.3±0.1** |

**Results under Mixed Distribution Shifts.** We evaluate different methods on the mixture of 15 corruption types (total of 15×50,000 images) at different severity levels (5&3). From Table 3, our SAR performs best consistently regarding accuracy, suggesting its effectiveness. Tent fails (occurs collapse) on ResNet50-GN levels 5&3 and VitBase-LN level 5 and thus achieves inferior accuracy than no adapt model, showing the instability of long-range online entropy minimization. Compared with Tent, although MEMO and DDA achieve better results, they rely on much more computation (inefficient at test time) as in Table 1, and DDA also needs to alter the training process (diffusion model training). By comparing Tent and EATA (both of them are efficient entropy-based), EATA achieves good results but it needs to pre-collect a set of in-distribution test samples (2,000 images in EATA) to compute Fisher importance for regularization and then adapt, which sometimes may be infeasible in practice. Unlike EATA, our SAR does not need such samples and obtains better results than EATA.

Table 3: Comparisons with state-of-the-arts on ImageNet-C under **MIXTURE OF 15 CORRUPTION TYPES** regarding **Accuracy (%)**.

| Model + Method | Level 5 | Level 3 |
|---|---|---|
| ResNet50 (BN) | 18.0 | 39.7 |
| • MEMO (Zhang et al., 2022) | 23.9 | 46.2 |
| • DDA (Gao et al., 2022) | 27.3 | 44.2 |
| • Tent (Wang et al., 2021) | 2.3 | 41.1 |
| • EATA (Niu et al., 2022a) | 26.8 | 52.6 |
| ResNet50 (GN) | 30.6 | 54.0 |
| • MEMO (Zhang et al., 2022) | 31.2 | 54.5 |
| • DDA (Gao et al., 2022) | 35.1 | 52.3 |
| • Tent (Wang et al., 2021) | 13.4 | 33.1 |
| • EATA (Niu et al., 2022a) | 38.1 | 56.1 |
| • SAR (ours) | **38.3±0.1** | **57.4±0.1** |
| VitBase (LN) | 29.9 | 53.8 |
| • MEMO (Zhang et al., 2022) | 39.1 | 62.1 |
| • DDA (Gao et al., 2022) | 36.1 | 53.2 |
| • Tent (Wang et al., 2021) | 16.5 | 70.2 |
| • EATA (Niu et al., 2022a) | 55.7 | 69.6 |
| • SAR (ours) | **57.1±0.1** | **70.7±0.1** |

**Results under Batch Size = 1.** From Table 4, our SAR achieves the best results in many cases. It is worth noting that MEMO and DDA are not affected by small batch sizes, mix domain shifts, or online imbalanced label shifts. They achieve stable/same results under these settings since they reset (or fix) the model parameters after the adaptation of each sample. However, the computational complexity of these two methods is much higher than SAR (see Table 1) and only obtain limited performance gains since they cannot exploit the knowledge from previously seen images. Although EATA performs better than our SAR on ResNet50-GN, it relies on pre-collecting 2,000 additional in-distribution samples (while we do not). Moreover, our SAR consistently outperforms EATA in other cases, see batch size 1 results on VitBase-LN, Tables 2-3, and Tables 8-9 in Appendix.

Table 5: Effects of components in SAR. We report the **Accuracy (%)** on ImageNet-C (level 5) under **ONLINE IMBALANCED LABEL SHIFTS** (imbalance ratio $q_{max}/q_{min} = \infty$). "reliable" and "sharpness-aware (sa)" denote Eqn. (2) and Eqn. (3), "recover" denotes the model recovery scheme.

| | Noise | | | Blur | | | | Weather | | | | Digital | | | | |
| Model+Method | Gauss. | Shot | Impul. | Defoc. | Glass | Motion | Zoom | Snow | Frost | Fog | Brit. | Contr. | Elastic | Pixel | JPEG | Avg. |
|---|---|---|---|---|---|---|---|---|---|---|---|---|---|---|---|---|
| ResNet50 (GN)+Entropy | 3.2 | 4.1 | 4.0 | 17.1 | 8.5 | 27.0 | 24.4 | 17.9 | 25.5 | 2.6 | 72.1 | 45.8 | 8.2 | 52.2 | 56.2 | 24.6 |
| ✦ reliable | 34.5 | 36.8 | 36.2 | 19.5 | 3.1 | 33.6 | 14.5 | 20.5 | 38.3 | 2.4 | 71.9 | 47.0 | 8.3 | 52.1 | 56.4 | 31.7 |
| ✦ reliable+sa | 33.8 | 35.9 | 36.4 | 19.2 | 18.7 | 33.6 | 24.5 | 23.5 | 45.2 | 49.3 | 71.9 | 46.6 | 9.2 | 51.6 | 56.4 | **37.0** |
| ✦ reliable+sa+recover | 33.6 | 36.1 | 36.2 | 19.1 | 18.6 | 33.9 | 24.7 | 22.5 | 45.7 | 49.0 | 71.9 | 46.6 | 9.2 | 51.5 | 56.3 | **37.0** |
| VitBase (LN)+Entropy | 21.2 | 1.9 | 38.6 | 54.8 | 52.7 | 58.5 | 54.2 | 10.1 | 14.7 | 69.6 | 76.3 | 66.3 | 59.2 | 69.7 | 66.8 | 47.6 |
| ✦ reliable | 47.8 | 35.7 | 48.4 | 55.2 | 54.1 | 58.6 | 54.4 | 13.3 | 21.4 | 69.5 | 76.2 | 66.1 | 60.2 | 69.3 | 66.7 | 53.1 |
| ✦ reliable+sa | 47.9 | 47.6 | 48.5 | 55.4 | 54.2 | 58.8 | 54.6 | 19.7 | 22.1 | 69.4 | 76.3 | 66.2 | 60.9 | 69.4 | 66.6 | 54.5 |
| ✦ reliable+sa+recover | 47.9 | 47.6 | 48.5 | 55.4 | 54.2 | 58.8 | 54.6 | 49.1 | 48.3 | 69.4 | 76.3 | 66.2 | 60.9 | 69.4 | 66.6 | **58.2** |

## 5.2 ABLATE EXPERIMENTS

**Comparison with Gradient Clipping.** As mentioned in Section 3.2, gradient clipping is a straightforward solution to alleviate model collapse. Here, we compare our SAR with two variants of gradient clip, *i.e.*, by value and by norm. From Figure 6, for both two variants, it is hard to set a proper threshold $\delta$ for clipping, since the gradients for different models and test data have different scales and thus the $\delta$ selection would

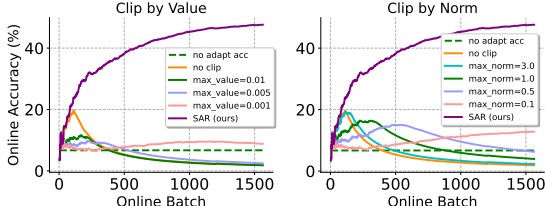

Figure 6: Comparison with gradient clipping. Results on VitBase-LN, ImageNet-C, shot noise, severity level 5, online imbalanced (ratio = $\infty$) label shift. *Accuracy is calculated over all previous test samples.*

be sensitive. We carefully select $\delta$ on a specific test set (*shot noise* level 5). Then, we select a very small $\delta$ to make gradient clip work, *i.e.*, clip by value 0.001 and by norm 0.1. Nonetheless, the performance gain over "no adapt" is very marginal, since the small $\delta$ would limit the learning ability of the model and in this case the clipped gradients may point in a very different direction from the true gradients. However, a large $\delta$ fails to stabilize the adaptation process and the accuracy will degrade after the model collapses (*e.g.*, clip by value 0.005 and by norm 1.0). In contrast, SAR does not need to tune such a parameter and achieve significant improvements than gradient clipping.

**Effects of Components in SAR.** From Table 5, compared with pure entropy minimization, the reliable entropy in Eqn. (2) clearly improves the adaptation performance, *i.e.*, $47.6\% \rightarrow 53.1\%$ on VitBase-LN and $24.6\% \rightarrow 31.7\%$ on ResNet50-GN w.r.t. average accuracy. With sharpness-aware (sa) minimization in Eqn. (3), the accuracy is further improved, *e.g.*, $31.7\% \rightarrow 37.0\%$ on ResNet50-GN w.r.t. average accuracy. On VitBase-LN, the sa module is also effective, and it helps the model to avoid collapse, *e.g.*, $35.7\% \rightarrow 47.6\%$ on *shot noise*. With both reliable and sa, our method performs stably except for very few cases, *i.e.*, VitBase-LN on *snow* and *frost*. For this case, our model recovery scheme takes effects, *i.e.*, $54.5\% \rightarrow 58.2\%$ on VitBase-LN w.r.t. average accuracy.

**Sharpness of Loss Surface.** We visualize the loss surface by adding perturbations to model weights, as done in Li et al. (2018b). We plot Figure 7 via the model weights obtained after the adaptation on the whole test set. By comparing Figures 7 (a) and (b), the area (the deepest blue) within the lowest loss contour line of our SAR is larger than Tent, showing that our solution is flatter and thus is more robust to noisy/large gradients.

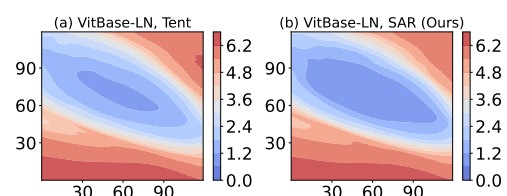

Figure 7: Loss (entropy) surface. Models are learned on ImageNet-C Gaussian noise, level 5.

## 6 CONCLUSIONS

In this paper, we seek to stabilize online test-time adaptation (TTA) under wild test settings, *i.e.*, mix shifts, small batch, and imbalanced label shifts. To this end, we first analyze and conduct extensive empirical studies to verify why wild TTA fails. Then, we point out that batch norm acts as a crucial obstacle to stable TTA. Meanwhile, though batch-agnostic norm (*i.e.*, group and layer norm) performs more stably under wild settings, they still suffer from many failure cases. To address these failures, we propose a sharpness-aware and reliable entropy minimization method (SAR) by suppressing the effect of certain noisy test samples with large gradients. Extensive experimental results demonstrate the stability and efficiency of our SAR under wild test settings.

## ACKNOWLEDGMENTS

This work was partially supported by the Key Realm R&D Program of Guangzhou 202007030007, National Natural Science Foundation of China (NSFC) 62072190, Ministry of Science and Technology Foundation Project 2020AAA0106900, Program for Guangdong Introducing Innovative and Enterpreneurial Teams 2017ZT07X183, CCF-Tencent Open Fund RAGR20220108.

## REPRODUCIBILITY STATEMENT

In this work, we implement all methods (all compared methods and our SAR) with different models (ResNet50-BN, ResNet50-GN, VitBase-LN) on the ImageNet-C/R and VisDA-2021 datasets. Reproducing all the results in our paper depends on the following three aspects:

1. **DATASET.** The first paragraph of Section 5 and Appendix C.1 provide the details of the adopted datasets and the download url.
2. **MODELS.** All adopted models (with the pre-trained weights) for test-time adaptation are publicly available. Specifically, ResNet50-BN is from torchvision, ResNet50-GN and VitBase-LN are from timm repository (Wightman, 2019). Appendix C.2 provides the download url of them.
3. **PROTOCOLS OF EACH METHOD.** The second paragraph of Section 5 and Appendix C.2 provides the implementation details of all compared methods and our SAR. We reproduce all compared methods based on the code from their official GitHub, for which the download url is provided (in Appendix C.2) following each method introduction. The source code of SAR has been made publicly available.

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

# APPENDIX

## CONTENTS

# A    RELATED WORK

We relate our SAR to existing adaptation methods without and with target data, sharpness-aware optimization, online learning methods, and EATA (Niu et al., 2022a).

**Adaptation without Target Data.** The problem of conquering distribution shifts has been studied in a number of works at *training time*, including domain generalization (Shankar et al., 2018; Li et al., 2018a; Dou et al., 2019), increasing the training dataset size (Orhan, 2019), various data augmentation techniques (Lim et al., 2019; Hendrycks et al., 2020; Li et al., 2021; Hendrycks et al., 2021; Yao et al., 2022), to name just a new. These methods aim to pre-anticipate or simulate the possible shifts of test data at training time, so that the training distribution can cover the possible shifts of test data. However, pre-anticipating all possible test shifts at training time may be infeasible and these training strategies are often more computationally expensive. Instead of improving generalization ability at training time, we conquer test shifts by directly learning from test data.

**Adaptation with Target Data.** We divide the discussion on related methods that exploit target data into 1) unsupervised domain adaptation (adapt offline) and 2) test-time adaptation (adapt online).

• *Unsupervised domain adaptation (UDA).* Conventional UDA jointly optimizes on the labeled source and unlabeled target data to mitigate distribution shifts, such as devising a domain discriminator to align source and target domains at feature level (Pei et al., 2018; Saito et al., 2018; Zhang et al., 2020b;a) and aligning the prototypes of source and target domains through a contrastive learning manner (Lin et al., 2022). Recently, *source-free UDA* methods have been proposed to resolve the adaptation problem when source data are absent, such as generative-based methods that generate source images or prototypes from the model (Li et al., 2020; Kundu et al., 2020; Qiu et al., 2021), and information maximization (Liang et al., 2020). These methods adapt models on a whole test set, in which the adaptation is offline and often requires multiple training epochs, and thus are hard to be deployed on online testing scenarios.

• *Test-time adaptation (TTA).* According to whether alter training, TTA methods can be mainly categorized into two groups. i) *Test-Time Training (TTT)* (Sun et al., 2020) jointly optimizes a source model with both supervised and self-supervised losses, and then conducts self-supervised learning at test time. The self-supervised losses can be rotation prediction (Gidaris et al., 2018) in TTT or contrastive-based objectives (Chen et al., 2020) in TTT++ (Liu et al., 2021) and MT3 (Bartler et al., 2022), *etc.* ii) *Fully Test-Time Adaptation* (Wang et al., 2021; Niu et al., 2022a; Hong et al., 2023) does not alter the training process and can be applied to any pre-trained model, including adapting the statistics in batch normalization layers (Schneider et al., 2020; Hu et al., 2021; Khurana et al., 2021; Lim et al., 2023; Zhao et al., 2023), unsupervised entropy minimization (Wang et al., 2021; Niu et al., 2022a; Zhang et al., 2022), prediction consistency maximization (Zhang et al., 2022; Wang et al., 2022; Chen et al., 2022a), top-$k$ classification boosting (Niu et al., 2022b), *etc.*

Though effective at handling test shifts, prior TTA methods are shown to be unstable in the online adaptation process and sensitive to when test data are insufficient (small batch sizes), from mixed domains, have imbalanced and online shifted label distribution (see Figure 1). Here, it is worth noting that methods like MEMO (Zhang et al., 2022) and DDA (Gao et al., 2022) are not affected under the above 3 scenarios, since MEMO resets the model parameters after each sample adaptation and DDA performs input adaptation via diffusion (in which the model weights are frozen during testing). However, these methods can not exploit the knowledge learned from previously seen samples and thus obtain limited performance gains. Moreover, the heavy data augmentations and diffusion in MEMO and DDA are computationally expensive and inefficient at test time (see Table 6). In this work, we analyze why online TTA may fail under the above practical test settings and propose associated solutions to make TTA stable under various wild test settings.

**Sharpness-aware Minimization (SAM).** SAM (Foret et al., 2021) optimizes both a supervised objective (*e.g.*, cross-entropy) and the sharpness of loss surface, aiming to find a flat minimum that has good generalization ability (Hochreiter & Schmidhuber, 1997). SAM and its variants (Kwon et al., 2021; Zheng et al., 2021; Du et al., 2022; Chen et al., 2022b) have shown outstanding performance on several deep learning benchmarks. In this work, when we analyze the failure reasons of test-time entropy minimization, we find that some noisy samples that produce gradients with large norms harm the adaptation and thus lead to model collapse. To alleviate this, we propose to minimize the

sharpness of the test-time entropy loss surface so that the online model update is robust to those noisy/large gradients.

**Online Learning (OL).** OL (Hoi et al., 2021; Chowdhury & Gopalan, 2017; Zhao et al., 2011) conducts model learning from a sequence of data samples one by one at a time, which is common in many real-world applications (*e.g.*, social web recommendation). According to the supervision type, OL can be categorized into three groups: i) *Supervised* methods (Rakhlin et al., 2010; Shalev-Shwartz et al., 2012) obtain supervision at the end of each online learning iteration, ii) *Semi-supervised* methods (Zhang & Hoi, 2019) can obtain supervision from only partial samples, *e.g.*, online active learning (Zhao & Hoi, 2013; Zhang et al., 2019) selects informative samples to query the ground-truth label for the model update, iii) *Unsupervised* methods (Bhatnagar et al., 2014) can not obtain any supervision during the whole online learning process. In this sense, test-time adaptation (TTA) (Sun et al., 2020; Wang et al., 2021) online updates models with only unlabeled test data and thus falls into the third category. However, unlike unsupervised OL that mainly aims to learn representations or clusters (Ren et al., 2021), TTA seeks to boost the performance of any pre-trained model on out-of-distribution test samples.

**Comparison with EATA (Niu et al., 2022a).** Although both EATA and our SAR include a step to remove samples via entropy, their motivations behind this step are different. EATA seeks to improve the adaptation efficiency via sample entropy selection. In our SAR, we discover that some noisy gradients with large norms may hurt the adaptation and thus result in model collapse under wild test settings. To remove these gradients, we exploit an alternative metric (*i.e.*, entropy), which helps to remove partial noisy gradients with large norms. However, this is still insufficient for achieving stable TTA (see ablation results in Table 5). Thus, we further introduce the sharpness-aware optimization and a model recovery scheme. With these three strategies, our SAR performs stably under wild test settings.

## B    PSEUDO CODE OF SAR

In this appendix, we provide the pseudo-code of our SAR method. From Algorithm 1, for each test sample $\mathbf{x}_j$, we first apply the reliable sample filtering scheme (refer to lines 3-6) to it to determine whether it will be used to update the model. If $\mathbf{x}_j$ is reliable, we will optimize the model via the sharpness-aware entropy loss of $\mathbf{x}_j$ (refer to lines 7-10). Specifically, we first calculate the optimal weight perturbation $\hat{\epsilon}(\tilde{\Theta})$ based on the gradient $\nabla_{\tilde{\Theta}}E(\mathbf{x}_j;\Theta)$, and then update the model with approximate gradients $\mathbf{g} = \nabla_{\tilde{\Theta}}E(\mathbf{x}_j;\Theta)|_{\Theta+\hat{\epsilon}(\tilde{\Theta})}$. Lastly, we exploit a recovery scheme to enable the model to work well even under a few extremely hard cases (refer to lines 11-13). Specifically, when the moving average value $e_m$ of entropy loss is smaller than $e_0$ (indicating that the model occurs to collapse), we will recover the model parameters $\tilde{\Theta}$ to its original/initial value.

---

**Algorithm 1: S**harpness-**A**ware and **R**eliable Test-Time Entropy Minimization (SAR)

---

**Input:** Test samples $\mathcal{D}_{test} = \{\mathbf{x}_j\}_{j=1}^M$, model $f_\Theta(\cdot)$ with trainable parameters $\tilde{\Theta} \subset \Theta$, step size $\eta > 0$, neighborhood size $\rho > 0$, $E_0 > 0$ in Eqn. (2), $e_0 > 0$ for model recovery.
**Output:** Predictions $\{\hat{y}_j\}_{j=1}^M$.

1  Initialize $\tilde{\Theta}_0 = \tilde{\Theta}$, moving average of entropy $e_m = 0$;
2  **for** $\mathbf{x}_j \in \mathcal{D}_{test}$ **do**
3       Compute entropy $E_j{=}E(\mathbf{x}_j;\Theta)$ and predict $\hat{y}_j{=}f_\Theta(\mathbf{x}_j)$;
4       **if** $E_j > E_0$ **then**
5           **continue** ;                          // reliable entropy minimization (Eqn. 2)
6       **end**
7       Compute gradient $\nabla_{\tilde{\Theta}}E(\mathbf{x}_j;\Theta)$;
8       Compute $\hat{\epsilon}(\tilde{\Theta})$ per Eqn. (4);
9       Compute gradient approximation: $\mathbf{g} = \nabla_{\tilde{\Theta}}E(\mathbf{x}_j;\Theta)|_{\Theta+\hat{\epsilon}(\tilde{\Theta})}$ ;
10      Update $\tilde{\Theta} \leftarrow \tilde{\Theta} - \eta\mathbf{g}$ ;                     // sharpness-aware minimization (Eqn. 3)
11      $e_m{=}0.9{\times}e_m{+}0.1{\times}E(\mathbf{x}_j;\Theta{+}\hat{\epsilon}(\tilde{\Theta}))$ **if** $e_m{\neq}0$ **else** $E(\mathbf{x}_j;\Theta{+}\hat{\epsilon}(\tilde{\Theta}))$ ;     // moving average
12      **if** $e_m < e_0$ **then**
13          Recover model weights: $\tilde{\Theta} \leftarrow \tilde{\Theta}_0$ ;                          // model recovery
14      **end**
15 **end**

---

## C   MORE IMPLEMENTATION DETAILS

### C.1   MORE DETAILS ON DATASET

In this paper, we mainly evaluate the out-of-distribution generalization ability of all methods on a large-scale and widely used benchmark, namely **ImageNet-C**[1] (Hendrycks & Dietterich, 2019). ImageNet-C is constructed by corrupting the original ImageNet (Deng et al., 2009) test set. The corruption (as shown in Figure 8) consists of 15 different types, *i.e.*, Gaussian noise, shot noise, impulse noise, defocus blur, glass blue, motion blur, zoom blur, snow, frost, fog, brightness, contrast, elastic transformation, pixelation, and JPEG compression, in which each corruption type has 5 different severity levels and the larger severity level means more severe distribution shift. Then, we further conduct experiments on **ImageNet-R** (Hendrycks et al., 2021) and **VisDA-2021** (Bashkirova et al., 2022) to verify the effectiveness of our method. ImageNet-R contains 30,000 images with various artistic renditions of 200 ImageNet classes, which are primarily collected from Flickr and filtered by Amazon MTurk annotators. VisDA-2021 collects images from ImageNet-O/R/C and ObjectNet (Barbu et al., 2019). The domain shifts in VisDA-2021 include the changes in artistic visual styles, textures, viewpoints and corruptions.

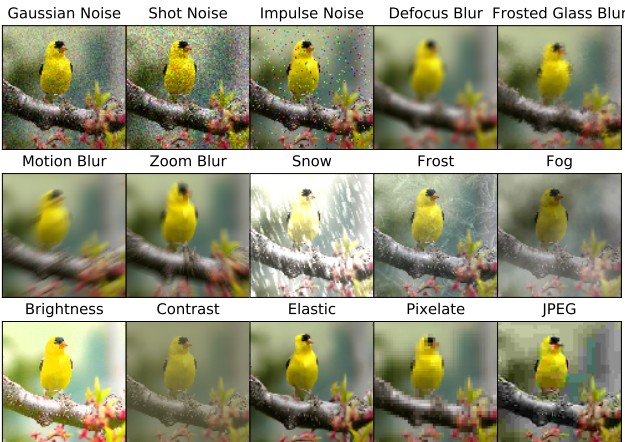

Figure 8: Visualizations of different corruption types in ImageNet corruption benchmark, which are taken from the original paper of ImageNet-C (Hendrycks & Dietterich, 2019).

### C.2   MORE EXPERIMENTAL PROTOCOLS

All pre-trained models involved in our paper for test-time adaptation are publicly available, including ResNet50-BN[2] obtained from `torchvision` library, ResNet-50-GN[3] and VitBase-LN[4] and ConvNeXt-LN[5] obtained from `timm` repository (Wightman, 2019). We summarize the detailed characteristics of all involved methods in Table 6 and introduce their implementation details in the following.

**SAR (Ours).** We use SGD as the update rule, with a momentum of 0.9, batch size of 64 (except for the experiments of batch size = 1), and learning rate of 0.00025/0.001 for ResNet/Vit models. The learning rate for batch size = 1 is set to (0.00025/16) for ResNet models and (0.001/32) for Vit models. The threshold $E_0$ in Eqn. (2) is set to $0.4 \times \ln 1000$ by following EATA (Niu et al., 2022a). $\rho$ in Eqn. (3) is set by the default value 0.05 in Foret et al. (2021). For model recovery, we record the

---

[1] https://zenodo.org/record/2235448#.YzQpq-xBxcA
[2] https://download.pytorch.org/models/resnet50-19c8e357.pth
[3] https://github.com/rwightman/pytorch-image-models/releases/download/v0.1-rsb-weights/resnet50_gn_a1h2-8fe6c4d0.pth
[4] https://storage.googleapis.com/vit_models/augreg/B_16-i21k-300ep-1r_0.001-aug_medium1-wd_0.1-do_0.0-sd_0.0--imagenet2012-steps_20k-1r_0.01-res_224.npz
[5] https://dl.fbaipublicfiles.com/convnext/convnext_base_1k_224_ema.pth

Table 6: Characteristics of state-of-the-art methods. We evaluate the efficiency of different methods with ResNet-50 (group norm) on ImageNet-C (Gaussian noise, severity level 5), which consists of totally 50,000 images. The real run time is tested via a single V100 GPU. DDA (Gao et al., 2022) pre-trains an additional diffusion model and then perform input adaptation/diffusion at test time.

| Method | Need source data? | Online update? | #Forward | #Backward | Other computation | GPU time (50,000 images) |
|---|---|---|---|---|---|---|
| MEMO (Zhang et al., 2022) | ✗ | ✗ | 50,000×65 | 50,000×64 | AugMix (Hendrycks et al., 2020) | 55,980 seconds |
| DDA (Gao et al., 2022) | ✓ | ✗ | 50,000×2 | 0 | 50,000 diffusion | 146,220 seconds |
| TTT (Sun et al., 2020) | ✓ | ✓ | 50,000×21 | 50,000×20 | rotation augmentation | 3,600 seconds |
| Tent (Wang et al., 2021) | ✗ | ✓ | 50,000 | 50,000 | n/a | 110 seconds |
| EATA (Niu et al., 2022a) | ✓ | ✓ | 50,000 | 26,196 | regularizer | 114 seconds |
| SAR (ours) | ✗ | ✓ | 50,000 + 12,710 | 12,710×2 | Eqn. (4) | 115 seconds |

entropy loss values with a moving average factor of 0.9 for $e_m$, and the reset threshold $e_0$ is set to 0.2. For **learnable parameters**, we only update affine parameters in normalization layers by following Tent (Wang et al., 2021). However, since the top/deep layers are more sensitive and more important to the original model than shallow layers as mentioned in (Mummadi et al., 2021; Choi et al., 2022), we freeze the top layers and update the affine parameters of layer or group normalization in the remaining shallow layers. Specifically, for ResNet50-GN that has 4 layer groups (layer1, 2, 3, 4), we freeze the layer4. For ViTBase-LN that has 11 blocks groups (blocks1-11), we freeze blocks9, blocks10, blocks11.

**TTT**[6] **(Sun et al., 2020).** For fair comparisons, we seek to compare all methods based on the same model weights. However, TTT alters the model training process and requires the model contains a self-supervised rotation prediction branch for test-time training. Therefore, we modify TTT so that it can be applied to any pre-trained model. Specifically, given a pre-trained model, we add a new branch (random initialized) from the end of a middle layer (2nd layer group of ResNet-50-GN and 6th blocks group of VitBase-LN) for the rotation prediction task. We first freeze all original parameters of the pre-trained model and train the newly added branch for 10 epochs on the original ImageNet training set. Here, we apply an SGD optimizer, with a momentum of 0.9, an initial learning rate of 0.1/0.005 for ResNet50-GN/VitBase-LN, and decrease it at epochs 4 and 7 by decreasing factor 0.1. Then, we take the newly obtained model (with two branches) as the base model to perform test-time training. During the test-time training phase, we use SGD as the update rule with a learning rate of 0.001 for ResNet0-GN (following TTT) and 0.0001 for VitBase-LN, and the data augmentation size is set to 20 (following Niu et al. (2022a)).

**Tent**[7] **(Wang et al., 2021).** We follow all hyper-parameters that are set in Tent unless it does not provide. Specifically, we use SGD as the update rule, with a momentum of 0.9, batch size of 64 (except for the experiments of batch size = 1 and effects of small test batch sizes (in Section 4)), and learning rate of 0.00025/0.001 for ResNet/Vit models. The learning rate for batch size = 1 is set to (0.00025/32) for ResNet models and (0.001/64) for Vit models. The trainable parameters are all affine parameters of batch normalization layers.

**EATA**[8] **(Niu et al., 2022a).** We follow all hyper-parameters that are set in EATA unless it does not provide. Specifically, the entropy constant $E_0$ (for reliable sample identification) is set to $0.1 \times \ln 1000$. The $\epsilon$ for redundant sample identification is set to 0.05. The trade-off parameter $\beta$ for entropy loss and regularization loss is set to 2,000. The number of pre-collected in-distribution test samples for Fisher importance calculation is 2,000. The update rule is SGD, with a momentum of 0.9, batch size of 64 (except for the experiments of batch size = 1 and effects of small test batch sizes (in Section 4)), and learning rate of 0.00025/0.001 for ResNet/Vit models. The learning rate for batch size = 1 is set to (0.00025/32) for ResNet models and (0.001/64) for Vit models. The trainable parameters are all affine parameters of batch normalization layers.

**MEMO**[9] **(Zhang et al., 2022).** We follow all hyper-parameters that are set in MEMO. Specifically, we use the AugMix (Hendrycks et al., 2020) as a set of data augmentations and the augmentation size is set to 64. For Vit models, the optimizer is AdamW (Loshchilov & Hutter, 2018), with learning

---

[6] https://github.com/yueatsprograms/ttt_imagenet_release
[7] https://github.com/DequanWang/tent
[8] https://github.com/mr-eggplant/EATA
[9] https://github.com/zhangmarvin/memo

rate 0.00001 and weight decay 0.01. For ResNet models, the optimizer is SGD, with learning rate 0.00025 and no weight decay. The trainable parameters are the entire model.

**DDA**[10] **(Gao et al., 2022).** We reproduce DDA according to its official GitHub repository and use the default hyper-parameters.

**More Details on Experiments in Section 4: Normalization Layer Effects in TTA.** In Section 4, we investigate the effects of TTT and Tent with models that have different norm layers under {small test batch sizes, mixed distribution shifts, online imbalanced label distribution shifts}. For each experiment, we only consider one of the three above test settings. To be specific, for experiments regarding batch size effects (Section 4 (1)), we only tune the batch size and the test set does not contain multiple types of distribution shifts and its label distribution is always uniform. For experiments of mixed domain shifts (Section 4 (2)), the test samples come from the mixture of 15 corruption types, while the batch size is 64 for Tent and 1 for TTT, and the label distribution of test data is always uniform. For experiments of online label shifts (Section 4 (3)), the label distribution of test data is online shifted and imbalanced, while the BS is 64 for Tent and 1 for TTT, and test data only consist of one corruption type. Moreover, it is worth noting that we re-scale the learning rate for entropy minimization (Tent) according to the batch size, since entropy minimization is sensitive to the learning rate and a fixed learning rate often fails to work well. Specifically, the learning rate is re-scaled as $(0.00025/32) \times \text{BS}$ IF $\text{BS} < 32$ ELSE $0.00025$ for ResNet models and $(0.001/64) \times \text{BS}$ for Vit models. Compared with Tent, the single sample adaptation method TTT is not very sensitive to the learning rate, and thus we set the same learning rate for various batch sizes. We also provide the results of TTT under different batch sizes with dynamic re-scaled learning rates in Table 7.

Table 7: Batch size (BS) effects in TTT (Sun et al., 2020) with different models (different norm layers). The learning rate is dynamically re-scaled by $0.001 \times \text{BS}$. We report the accuracy (%) on ImageNet-C with Gaussian noise and severity level 5.

| Model | BS=1 | BS=2 | BS=4 | BS=8 | BS=16 |
|---|---|---|---|---|---|
| ResNet50-BN | 21.2 | 23.4 | 23.4 | 24.7 | 24.6 |
| ResNet50-GN | 40.9 | 40.5 | 40.8 | 41.1 | 40.7 |

---

[10]https://github.com/shiyegao/DDA

# D  ADDITIONAL RESULTS ON IMAGENET-C OF SEVERITY LEVEL 3

## D.1  COMPARISONS WITH STATE-OF-THE-ARTS UNDER ONLINE IMBALANCED LABEL SHIFT

We provide more results regarding online imbalanced label distribution shift (imbalance ratio $= \infty$) of all compared methods in Table 8. The results are consistent with that of the main paper (severity level 5), and our SAR performs best in the average of 15 different corruption types. It is worth noting that DDA achieves competitive results under *noise* corruptions while performing worse for other corruption types. The reason is that the diffusion model used in DDA for input adaptation is trained via noise diffusion, and thus its generalization ability to diffuse other corruptions is still limited.

Table 8: Comparisons with state-of-the-art methods on ImageNet-C of severity level 3 under **ONLINE IMBALANCED LABEL DISTRIBUTION SHIFTS** (imbalance ratio $q_{max}/q_{min} = \infty$) regarding **Accuracy (%)**. "BN"/"GN"/"LN" is short for Batch/Group/Layer normalization.

| | Noise | | | Blur | | | | Weather | | | | Digital | | | | |
|---|---|---|---|---|---|---|---|---|---|---|---|---|---|---|---|---|
| Model+Method | Gauss. | Shot | Impul. | Defoc. | Glass | Motion | Zoom | Snow | Frost | Fog | Brit. | Contr. | Elastic | Pixel | JPEG | Avg. |
| ResNet50 (BN) | 27.7 | 25.2 | 25.1 | 37.8 | 16.7 | 37.8 | 35.3 | 35.2 | 32.1 | 46.7 | 69.5 | 46.2 | 55.4 | 46.2 | 59.4 | 39.8 |
| • MEMO | 37.6 | 34.5 | 36.7 | 41.4 | 23.4 | 44.4 | 40.9 | 44.6 | 37.3 | 52.4 | 70.5 | 56.3 | 58.7 | 55.2 | 60.9 | 46.3 |
| • DDA | 49.9 | 50.0 | 49.2 | 33.2 | 31.9 | 38.0 | 36.7 | 35.1 | 34.1 | 35.01 | 64.9 | 33.7 | 59.3 | 53.9 | 59.0 | 44.3 |
| • Tent | 3.4 | 3.2 | 3.2 | 2.3 | 2.0 | 2.4 | 3.4 | 2.4 | 2.4 | 4.6 | 5.4 | 3.0 | 4.8 | 4.6 | 4.5 | 3.4 |
| • EATA | 1.3 | 0.9 | 1.1 | 0.6 | 0.6 | 1.2 | 1.4 | 1.3 | 1.3 | 1.9 | 4.1 | 1.6 | 2.7 | 2.4 | 3.1 | 1.7 |
| ResNet50 (GN) | 54.5 | 52.9 | 53.1 | 44.4 | 21.2 | 49.8 | 39.3 | 54.9 | 54.1 | 55.8 | 75.3 | 69.7 | 59.6 | 59.7 | 66.4 | 54.1 |
| • MEMO | 55.9 | 54.3 | 54.1 | 40.1 | 23.1 | 49.5 | 41.4 | 54.8 | 54.1 | 57.6 | 75.7 | 70.2 | 60.2 | 61.5 | 66.7 | 54.6 |
| • DDA | 61.0 | 61.0 | 60.5 | 39.3 | 37.3 | 46.4 | 39.7 | 47.7 | 48.1 | 29.9 | 69.9 | 57.8 | 62.8 | 60.1 | 63.8 | 52.4 |
| • Tent | 59.1 | 58.6 | 58.3 | 39.0 | 27.9 | 54.7 | 41.1 | 51.3 | 41.4 | 62.0 | 75.2 | 70.1 | 62.3 | 63.7 | 66.4 | 55.4 |
| • EATA | 52.3 | 52.9 | 51.7 | 35.7 | 30.1 | 46.4 | 39.6 | 43.8 | 39.8 | 55.7 | 72.4 | 66.6 | 54.7 | 56.0 | 56.2 | 50.3 |
| • SAR (ours) | 60.8±0.1 | 60.5±0.3 | 60.2±0.2 | 47.9±0.5 | 36.7±0.7 | 58.2±0.2 | 49.7±0.5 | 57.9±0.3 | 53.6±0.0 | 65.0±0.1 | 76.4±0.2 | 71.0±0.0 | 67.0±0.2 | 65.8±0.1 | 67.6±0.0 | 59.9±0.1 |
| VitBase (LN) | 51.5 | 46.8 | 50.4 | 48.7 | 37.1 | 54.7 | 41.6 | 35.1 | 33.3 | 68.0 | 69.3 | 74.9 | 65.9 | 66.0 | 63.6 | 53.8 |
| • MEMO | 62.1 | 57.9 | 61.5 | 57.2 | 45.6 | 62.0 | 49.9 | 46.5 | 43.1 | 74.1 | 75.8 | 79.7 | 72.6 | 72.3 | 70.6 | 62.1 |
| • DDA | 59.7 | 58.2 | 59.4 | 43.5 | 43.3 | 50.5 | 41.0 | 34.3 | 34.4 | 55.4 | 65.0 | 64.2 | 64.1 | 63.8 | 62.9 | 53.3 |
| • Tent | 68.7 | 68.0 | 68.1 | 68.2 | 63.8 | 70.9 | 63.8 | 67.6 | 41.9 | 76.3 | 78.8 | 79.5 | 75.9 | 76.7 | 73.7 | 69.5 |
| • EATA | 65.3 | 62.6 | 63.6 | 63.0 | 57.1 | 66.3 | 59.3 | 64.5 | 61.0 | 73.3 | 76.9 | 75.9 | 74.2 | 74.8 | 73.1 | 67.4 |
| • SAR (ours) | 68.8±0.1 | 68.2±0.1 | 68.4±0.2 | 68.3±0.2 | 64.7±0.0 | 71.0±0.2 | 64.2±0.3 | 68.1±0.1 | 66.0±0.1 | 76.4±0.1 | 79.0±0.1 | 79.6±0.1 | 76.2±0.3 | 77.1±0.1 | 74.1±0.2 | 71.3±0.1 |

## D.2  COMPARISONS WITH STATE-OF-THE-ARTS UNDER BATCH SIZE OF 1

We provide more results regarding batch size = 1 of all compared methods in Table 9. The results are consistent with that of the main paper (severity level 5), and our SAR performs best in the average of 15 different corruption types.

Table 9: Comparisons with state-of-the-art methods on ImageNet-C of severity level 3 under **BATCH SIZE=1** regarding **Accuracy (%)**. "BN"/"GN"/"LN" is short for Batch/Group/Layer normalization.

| | Noise | | | Blur | | | | Weather | | | | Digital | | | | |
|---|---|---|---|---|---|---|---|---|---|---|---|---|---|---|---|---|
| Model+Method | Gauss. | Shot | Impul. | Defoc. | Glass | Motion | Zoom | Snow | Frost | Fog | Brit. | Contr. | Elastic | Pixel | JPEG | Avg. |
| ResNet50 (BN) | 27.6 | 25.0 | 25.1 | 38.0 | 16.9 | 37.7 | 35.2 | 35.2 | 32.1 | 46.6 | 69.6 | 46.0 | 55.6 | 46.2 | 59.3 | 39.7 |
| • MEMO | 37.5 | 34.3 | 36.6 | 41.2 | 23.3 | 44.2 | 41.0 | 44.5 | 37.4 | 52.3 | 70.5 | 56.0 | 56.7 | 55.0 | 60.8 | 46.2 |
| • DDA | 49.8 | 49.9 | 49.2 | 33.2 | 32.0 | 37.9 | 36.6 | 35.2 | 34.2 | 34.9 | 64.9 | 33.5 | 59.3 | 53.9 | 59.0 | 44.2 |
| • Tent | 0.1 | 0.2 | 0.2 | 0.2 | 0.1 | 0.1 | 0.2 | 0.2 | 0.2 | 0.2 | 0.2 | 0.2 | 0.2 | 0.2 | 0.2 | 0.2 |
| • EATA | 0.2 | 0.2 | 0.1 | 0.2 | 0.1 | 0.2 | 0.2 | 0.1 | 0.2 | 0.2 | 0.2 | 0.2 | 0.2 | 0.2 | 0.2 | 0.2 |
| ResNet50 (GN) | 54.5 | 52.8 | 53.1 | 44.3 | 21.2 | 49.7 | 39.2 | 54.8 | 54.0 | 55.8 | 75.4 | 69.8 | 59.6 | 59.7 | 66.3 | 54.0 |
| • MEMO | 55.7 | 54.2 | 53.9 | 40.0 | 22.8 | 49.2 | 41.2 | 54.8 | 54.1 | 57.6 | 75.5 | 69.9 | 60.0 | 61.3 | 66.6 | 54.5 |
| • DDA | 61.0 | 60.9 | 60.4 | 39.2 | 37.2 | 46.4 | 39.7 | 47.7 | 48.0 | 29.8 | 70.0 | 58.0 | 62.7 | 60.2 | 63.8 | 52.3 |
| • Tent | 58.8 | 58.5 | 58.7 | 38.2 | 26.8 | 54.9 | 42.6 | 51.6 | 38.8 | 61.9 | 75.3 | 70.0 | 62.3 | 63.6 | 66.3 | 55.2 |
| • EATA | 59.2 | 58.7 | 58.8 | 45.7 | 32.6 | 55.5 | 45.9 | 56.4 | 52.7 | 63.6 | 75.9 | 71.1 | 64.7 | 64.5 | 67.8 | 58.2 |
| • SAR (ours) | 60.3±0.1 | 59.6±0.1 | 59.5±0.1 | 46.6±1.1 | 33.0±0.5 | 57.5±0.1 | 47.8±0.1 | 57.8±0.2 | 52.8±0.1 | 65.1±0.1 | 76.7±0.1 | 71.4±0.1 | 67.3±0.2 | 66.0±0.1 | 67.8±0.0 | 59.3±0.1 |
| VitBase (LN) | 51.6 | 46.9 | 50.5 | 48.7 | 37.2 | 54.7 | 41.6 | 35.1 | 33.5 | 67.8 | 69.3 | 74.8 | 65.8 | 66.0 | 63.7 | 53.8 |
| • MEMO | 61.9 | 57.7 | 61.4 | 57.0 | 45.4 | 61.8 | 49.8 | 46.6 | 43.1 | 73.9 | 75.7 | 79.6 | 72.6 | 72.1 | 70.5 | 61.9 |
| • DDA | 59.8 | 58.2 | 59.5 | 43.4 | 43.2 | 50.4 | 40.9 | 34.2 | 34.3 | 55.2 | 64.9 | 64.0 | 64.2 | 63.7 | 62.8 | 53.2 |
| • Tent | 67.1 | 66.2 | 66.3 | 66.3 | 60.9 | 69.1 | 61.4 | 65.2 | 60.4 | 75.2 | 78.1 | 78.8 | 74.9 | 75.8 | 72.4 | 69.2 |
| • EATA | 60.7 | 58.5 | 61.6 | 60.1 | 51.8 | 64.2 | 54.8 | 53.3 | 52.6 | 72.5 | 73.6 | 77.9 | 71.3 | 71.3 | 69.7 | 63.6 |
| • SAR (ours) | 68.5±0.1 | 67.8±0.1 | 68.0±0.1 | 67.8±0.2 | 63.1±0.0 | 70.7±0.1 | 63.5±0.1 | 66.9±0.2 | 62.8±2.1 | 75.8±0.5 | 77.7±0.8 | 78.4±0.4 | 74.7±1.5 | 75.7±0.5 | 72.7±1.1 | 70.3±0.3 |

# E    ADDITIONAL RESULTS ON IMAGENET-R AND VISDA-2021

We further conduct experiments on ImageNet-R under two wild test settings: online imbalanced label distribution shifts (in Table 10) and batch size = 1 (in Table 11). The overall results are consistent with that on ImageNet-C: 1) ResNet50-GN and VitBase-LN perform more stable than ResNet50-BN; 2) Compared with Tent and EATA, SAR achieves the best performance on ResNet50-GN and VitBase-LN.

Table 10: Comparison in terms of accuracy (%) under the wild setting **online imbalanced label distribution shifts** on ImageNet-R.

| Method | ResNet50-BN | ResNet50-GN | VitBase-LN |
|---|---|---|---|
| No Adapt. | 36.2 | 40.8 | 43.1 |
| Tent | 6.6 | 41.7 | 45.2 |
| EATA | 5.8 | 40.9 | 47.5 |
| SAR (ours) | - | **42.9** | **52.0** |

Table 11: Comparison in terms of accuracy (%) under the wild setting **single sample adaptation (batch size = 1)** on ImageNet-R.

| Method | ResNet50-BN | ResNet50-GN | VitBase-LN |
|---|---|---|---|
| No Adapt. | 36.2 | 40.8 | 43.1 |
| Tent | 0.6 | 42.2 | 40.5 |
| EATA | 0.6 | 42.3 | 52.5 |
| SAR (ours) | - | **43.9** | **53.1** |

We also compare our method with online TTA methods (Tent and EATA) on VisDA-2021. Results in Table 12 are consistent with that on ImageNet-C and ImageNet-R. Specifically, Tent and EATA fail with the BN model (ResNet50-BN) while performing stable with the GN/LN model (ResNet50-GN/VitBase-LN) under online imbalanced label shifts. Besides, our SAR further improves the adaptation performance of Tent and EATA on ResNet50-GN/VitBase-LN.

Table 12: Classification **Accuracy (%)** on VisDA-2021 under **online imbalanced label distribution shifts** and **mixed domain shifts**.

| Model | No adapt | Tent | EATA | SAR (ours) |
|---|---|---|---|---|
| ResNet50-BN | 36.0 | 9.0 | 7.4 | – |
| ResNet50-GN | 43.7 | 42.8 | 40.9 | **44.1** |
| VitBase-LN | 44.3 | 49.1 | 49.0 | **52.0** |

## F  ADDITIONAL ABLATE RESULTS

### F.1  EFFECTS OF COMPONENTS IN SAR

We further ablate the effects of components in our SAR in Figure 9 by plotting the changes of gradients norms of our methods with different components. From the results, both the reliable (Eqn. 2) and sharpness-aware (sa) (Eqn. 3) modules together ensure the model's gradients keep in a normal range during the whole online adaptation process, which is consistent with our previous results in Section 5.2.

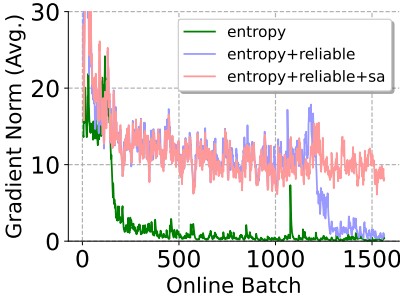

Figure 9: The evolution of gradients norm during online test-time adaptation. Results on VitBase, ImageNet-C, shot noise, severity level 5, online imbalance label shift (imbalance ratio = ∞). "reliable" and "sharpness-aware (sa)" are short for Eqn. (2) and Eqn. (3), respectively.

We also conduct more ablation experiments under the wild test settings of "batch size (BS)=1" in Table 13 and "mix domain shifts" in Table 14. The results are generally consistent with that in Table 5. Both the reliable entropy and sharpness-aware optimization work together to stabilize online TTA. It is worth noting that only in VitBase-LN under BS=1 the model recovery scheme is activated and improves the average accuracy from 55.7% to 56.4%.

Table 13: Effects of components in SAR. We report the **Accuracy (%)** on ImageNet-C (level 5) under **BATCH SIZE=1**. "reliable" and "sharpness-aware (sa)" denote Eqn. (2) and Eqn. (3), "recover" denotes the model recovery scheme.

| | Noise | | | Blur | | | | Weather | | | | Digital | | | | |
|---|---|---|---|---|---|---|---|---|---|---|---|---|---|---|---|---|
| Model+Method | Gauss. | Shot | Impul. | Defoc. | Glass | Motion | Zoom | Snow | Frost | Fog | Brit. | Contr. | Elastic | Pixel | JPEG | Avg. |
| ResNet50 (GN)+Entropy | 3.4 | 4.5 | 4.0 | 16.7 | 5.9 | 27.0 | 29.7 | 16.3 | 26.6 | 2.1 | 72.1 | 46.4 | 7.5 | 52.5 | 56.2 | 24.7 |
| ✛ reliable | 22.8 | 25.5 | 23.0 | 18.4 | 14.8 | 27.0 | 28.6 | 40.0 | 43.3 | 18.4 | 71.5 | 43.1 | 15.2 | 45.6 | 55.4 | 32.8 |
| ✛ reliable+sa | 23.8 | 26.4 | 24.0 | 18.6 | 15.4 | 28.3 | 30.5 | 44.8 | 44.8 | 26.7 | 72.4 | 44.5 | 12.2 | 46.9 | 65.1 | **34.4** |
| ✛ reliable+sa+recover | 23.8 | 26.4 | 23.9 | 18.5 | 15.4 | 28.3 | 30.6 | 44.6 | 44.9 | 24.7 | 72.4 | 44.4 | 12.3 | 47.0 | 56.1 | 34.2 |
| VitBase (LN)+Entropy | 42.9 | 1.4 | 43.9 | 52.6 | 48.8 | 55.8 | 51.4 | 22.2 | 20.1 | 67.0 | 75.1 | 64.7 | 53.7 | 67.2 | 64.5 | 48.8 |
| ✛ reliable | 34.8 | 4.2 | 35.5 | 50.5 | 45.9 | 54.0 | 48.6 | 52.5 | 47.8 | 65.5 | 74.5 | 63.4 | 51.4 | 65.6 | 63.0 | 50.5 |
| ✛ reliable+sa | 40.4 | 37.3 | 41.2 | 53.6 | 50.6 | 57.3 | 53.0 | 58.7 | 40.9 | 68.8 | 75.9 | 65.7 | 57.9 | 69.0 | 66.0 | 55.7 |
| ✛ reliable+sa+recover | 40.4 | 37.3 | 41.2 | 53.6 | 50.6 | 57.3 | 53.0 | 58.7 | 50.7 | 68.8 | 75.4 | 65.7 | 57.9 | 69.0 | 66.0 | **56.4** |

Table 14: Effects of components in SAR. We report the **Accuracy (%)** under **MIXED DOMAIN SHIFTS**, *i.e.*, mixture of 15 corruption types of ImageNet-C with severity level 5. "reliable" and "sharpness-aware (sa)" denote Eqn. (2) and Eqn. (3), "recover" denotes the model recovery scheme.

| Model + Method | Accuracy (%) |
|---|---|
| ResNet50 (GN)+Entropy | 33.5 |
| ✛ reliable | 38.1 |
| ✛ reliable+sa | 38.2 |
| ✛ reliable+sa+recover | 38.2 |
| ResNet50 (GN)+Entropy | 18.5 |
| ✛ reliable | 55.2 |
| ✛ reliable+sa | 57.2 |
| ✛ reliable+sa+recover | 57.2 |

### F.2 VISUALIZATION OF LOSS SURFACE LEARNED BY SAR

In Figure 7 in the main paper, we have visualized the loss surface of Tent and our SAR on VitBase-LN. In this section, we further provide visualizations of ResNet50-GN. We select a checkpoint at batch 120 to plot the loss surface for Tent, since after batch 120 this model starts to collapse. In this case, the loss (entropy) is hard to degrade and cannot find a proper minimum. For our SAR, the model weights for plotting are obtained after the adaptation on the whole test set. By comparing Figures 10 (a)-(b), SAR helps to stabilize the entropy minimization process and find proper minima.

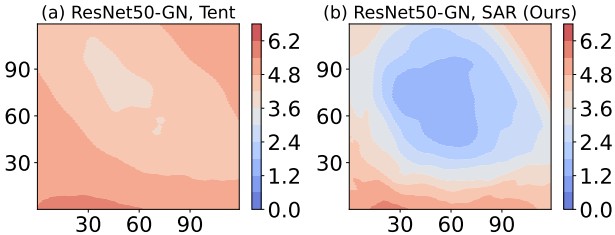

Figure 10: Visualization of loss (entropy) surface. Models are learned on ImageNet-C of Gaussian noise with severity level 5.

### F.3 SENSITIVITY OF $\rho$ IN SAR

The hyper-parameter $\rho$ (in Eqn. (3)) in sharpness-aware optimization is not hard to tune in our online test-time adaptation. Following Foret et al. (2021), we set $\rho = 0.05$ in all experiments, and it works well with different model architectures (ResNet50-BN, ResNet50-GN, VitBase-LN) on different datasets (ImageNet-C, ImageNet-R). Here, we also conduct a sensitivity analysis of $\rho$ in Table 15, in which SAR works well under the range [0.03, 0.1].

Table 15: Sensitivity of $\rho$ in SAR. We report **Accuracy (%)** on ImageNet-C (shot noise, severity level 5) **under online imbalanced label distribution shifts**, where the imbalance ratio is $\infty$.

| Method | No adapt | Tent | SAR ($\rho$=0.01) | SAR ($\rho$=0.03) | SAR ($\rho$=0.05) | SAR ($\rho$=0.07) | SAR ($\rho$=0.1) |
|---|---|---|---|---|---|---|---|
| Accuracy (%) | 6.7 | 1.4 | 40.3 | 47.7 | 47.6 | 47.2 | 47.2 |

## G    ADDITIONAL DISCUSSIONS

### G.1    MORE DISCUSSIONS ON MODEL COLLAPSE

As we mentioned in Section 3.1, models online optimized by entropy (Wang et al., 2021; Niu et al., 2022a) under wild test scenarios are easy to collapse, *i.e.*, predict all samples to a single class independent of the inputs. To alleviate this, prior methods (Liang et al., 2020; Mummadi et al., 2021) exploit diversity regularization to force the output distribution of samples to be uniform. However, this assumption is unreasonable at test time such as when test data are imbalanced during a period (as in Figure 1 (c)) and this method also relies on a batch of samples (in contrast to Figure 1 (b)). In this sense, this strategy is infeasible for our problem. In our paper, we resolve the collapse issue for test-time adaptation from an optimization perspective to make the online adaptation process stabilized.

### G.2    EFFECTS OF LARGE BATCH SIZES IN BN MODELS UNDER MIX DOMAIN SHIFTS

In Section 3.1, we mentioned that a standard batch size (*e.g.*, 64 on ImageNet) works well when there is only one type of distribution shift. However, when test data contains multiple shifts, this batch size fails to calculate an accurate mean and variance estimation in batch normalization layers. Here, we investigate the effects of super large batch sizes in this setting. From Table 16, the adapted performance increases as the batch size increases, indicating that a larger batch size helps to estimate statistics more accurately. It is worth noting that the performance on severity level 3 degrades when BS is larger than 1024. This is because we fix the learning rate for various batch sizes and in this sense, BS=1024 may lead to insufficient model updates. Moreover, although enlarging batch sizes is able to boost the performance, the adapt performance is still inferior to the average accuracy of adapting on each corruption type separately (*i.e.*, average adapt). This further emphasizes the necessity of exploiting models with group or layer norm layers to perform test-time entropy minimization.

Table 16: Effects of large batch sizes (BS) in Tent (Wang et al., 2021) with ResNet-50-BN under MIXTURE OF 15 DIFFERENT CORRUPTION TYPES on ImageNet-C. We report accuracy (%).

| Severity | Base | avg. adapt BS=64 | mix adapt BS=32 | BS=64 | BS=128 | BS=256 | BS=512 | BS=1,024 |
|---|---|---|---|---|---|---|---|---|
| Level = 5 | 18.0 | 42.6 | 0.9 | 2.3 | 4.0 | 8.3 | 12.4 | 16.4 |
| Level = 3 | 39.8 | 59.0 | 20.1 | 40.7 | 46.7 | 49.1 | 49.0 | 47.8 |

### G.3    PERFORMANCE OF TENT WITH CONVNEXT-LN

For mainstream neural network models, the normalization layers are often coupled with network architecture. Specifically, group norm (GN) and batch norm (BN) are often combined with conventional networks, while layer norm (LN) is more suitable for transformer networks. Therefore, we investigate the layer normalization effects in TTA in Section 4 through VitBase-LN. Here, we conduct more experiments to compare the performance of online entropy minimization (Wang et al., 2021) on ResNet50-BN and ConvNeXt-LN (Liu et al., 2022). ConvNeXt is a convolutional network equipped with LN. The authors conduct significant modifications over ResNet to make this LN-based convolutional network work well, such as modifying the architecture (ResNet block to ConvNeXt block, activation functions, etc.), various training strategies (stochastic depth, random erasing, EMA, etc.). From Table 17, Tent+ConvNeXt-LN performs more stable than Tent+ResNet50-BN, but still suffers several failure cases. These results are consistent with that of ResNet50-BN *vs.* VitBase-LN.

### G.4    EFFECTIVENESS OF MODEL RECOVERY SCHEME WITH TENT AND EATA

In this subsection, we apply our Model Recovery scheme to Tent (Wang et al., 2021) and EATA (Niu et al., 2022a). From Table 18, the model recovery indeed helps Tent a lot (*e.g.*, the average accuracy from 22.0% to 26.1% on ResNet50-GN) while its performance gain on EATA is a bit marginal. Compared with Tent+recovery and EATA+recovery, our SAR greatly boosts the adaptation performance,

Table 17: Results of Tent on ResNet50-BN and ConvNeXt-LN. We report **Accuracy (%)** on ImageNet-C under **online imbalanced label distribution shifts** and the imbalance ratio is $\infty$.

| Severity level 5 | Noise | | | Blur | | | | Weather | | | | Digital | | | | Avg. |
|---|---|---|---|---|---|---|---|---|---|---|---|---|---|---|---|---|
| | Gauss. | Shot | Impul. | Defoc. | Glass | Motion | Zoom | Snow | Frost | Fog | Brit. | Contr. | Elastic | Pixel | JPEG | |
| ResNet50-BN | 2.2 | 2.9 | 1.8 | 17.8 | 9.8 | 14.5 | 22.5 | 16.8 | 23.4 | 24.6 | 59.0 | 5.5 | 17.1 | 20.7 | 31.6 | 18.0 |
| • Tent (Wang et al., 2021) | 1.2 | 1.4 | 1.4 | 1.0 | 0.9 | 1.2 | 2.6 | 1.7 | 1.8 | 3.6 | 5.0 | 0.5 | 2.6 | 3.2 | 3.1 | 2.1 |
| ConvNeXt-LN | 52.3 | 52.7 | 52.3 | 31.7 | 18.7 | 42.5 | 38.1 | 54.2 | 58.3 | 50.6 | 75.6 | 56.8 | 32.2 | 39.2 | 60.4 | 47.7 |
| • Tent (Wang et al., 2021) | 26.3 | 11.9 | 36.9 | 31.1 | 12.7 | 14.6 | 5.1 | 7.8 | 5.3 | 6.6 | 79.0 | 67.6 | 1.5 | 68.4 | 65.8 | 29.4 |
| Severity level 3 | Gauss. | Shot | Impul. | Defoc. | Glass | Motion | Zoom | Snow | Frost | Fog | Brit. | Contr. | Elastic | Pixel | JPEG | Avg. |
| ResNet50-BN | 27.7 | 25.2 | 25.1 | 37.8 | 16.7 | 37.8 | 35.3 | 35.2 | 32.1 | 46.7 | 69.5 | 46.2 | 55.4 | 46.2 | 59.4 | 39.8 |
| • Tent (Wang et al., 2021) | 3.4 | 3.2 | 3.2 | 2.3 | 2.0 | 2.4 | 3.4 | 2.4 | 2.4 | 4.6 | 5.4 | 3.0 | 4.8 | 4.6 | 4.5 | 3.4 |
| ConvNeXt-LN | 71.1 | 69.8 | 72.4 | 55.2 | 36.6 | 64.3 | 52.8 | 63.2 | 64.0 | 66.0 | 79.4 | 75.8 | 69.1 | 68.5 | 71.9 | 65.4 |
| • Tent (Wang et al., 2021) | 73.4 | 73.5 | 74.4 | 66.8 | 61.6 | 71.4 | 64.2 | 22.6 | 39.5 | 76.8 | 81.2 | 78.9 | 76.8 | 75.6 | 75.8 | 67.5 |

*e.g.*, the average accuracy 26.1% (Tent+recovery) *vs.* 37.2% (SAR) on ResNet50-GN, suggesting the effectiveness of our proposed SAR.

Table 18: Results of combining model recovery scheme with Tent and EATA. We report the **Accuracy (%)** on ImageNet-C severity level 5 **under online imbalanced label distribution shifts** and the imbalance ratio is $\infty$.

| Model+Method | Noise | | | Blur | | | | Weather | | | | Digital | | | | Avg. |
|---|---|---|---|---|---|---|---|---|---|---|---|---|---|---|---|---|
| | Gauss. | Shot | Impul. | Defoc. | Glass | Motion | Zoom | Snow | Frost | Fog | Brit. | Contr. | Elastic | Pixel | JPEG | |
| ResNet50 (GN) | 18.0 | 19.8 | 17.9 | 19.8 | 11.4 | 21.4 | 24.9 | 40.4 | 47.3 | 33.6 | 69.3 | 36.3 | 18.6 | 28.4 | 52.3 | 30.6 |
| • Tent | 2.6 | 3.3 | 2.7 | 13.9 | 7.9 | 19.5 | 17.0 | 16.5 | 21.9 | 1.8 | 70.5 | 42.2 | 6.6 | 49.4 | 53.7 | 22.0 |
| • Tent+recover | 10.1 | 12.2 | 10.6 | 13.9 | 8.5 | 19.5 | 20.6 | 24.3 | 33.5 | 8.9 | 70.5 | 42.2 | 13.5 | 49.4 | 53.7 | 26.1 |
| • EATA | 27.0 | 28.3 | 28.1 | 14.9 | 17.1 | 24.4 | 25.3 | 32.2 | 32.0 | 39.8 | 66.7 | 33.6 | 24.5 | 41.9 | 38.4 | 31.6 |
| • EATA+recover | 26.1 | 31.0 | 27.2 | 19.9 | 18.5 | 25.7 | 25.7 | 35.9 | 28.6 | 40.4 | 68.2 | 35.3 | 27.6 | 42.9 | 40.9 | 32.9 |
| • SAR (ours) | 33.1 | 36.5 | 35.5 | 19.2 | 19.5 | 33.3 | 27.7 | 23.9 | 45.3 | 50.1 | 71.9 | 46.7 | 7.1 | 52.1 | 56.3 | **37.2** |
| VitBase (LN) | 9.4 | 6.7 | 8.3 | 29.1 | 23.4 | 34.0 | 27.0 | 15.8 | 26.3 | 47.4 | 54.7 | 43.9 | 30.5 | 44.5 | 47.6 | 29.9 |
| • Tent | 32.7 | 1.4 | 34.6 | 54.4 | 52.3 | 58.2 | 52.2 | 7.7 | 12.0 | 69.3 | 76.1 | 66.1 | 56.7 | 69.4 | 66.4 | 47.3 |
| • Tent+recover | 40.3 | 10.1 | 42.4 | 54.4 | 52.3 | 58.1 | 52.2 | 31.6 | 39.2 | 69.3 | 76.1 | 66.1 | 56.7 | 69.4 | 66.4 | 52.3 |
| • EATA | 35.9 | 34.6 | 36.7 | 45.3 | 47.2 | 49.3 | 47.7 | 56.5 | 55.4 | 62.2 | 72.2 | 21.7 | 56.2 | 64.7 | 63.7 | 50.0 |
| • EATA+recover | 35.9 | 34.6 | 36.7 | 45.3 | 47.2 | 49.3 | 47.7 | 56.5 | 55.4 | 62.2 | 72.2 | 21.7 | 56.2 | 64.7 | 63.7 | 49.9 |
| • SAR (ours) | 46.5 | 43.1 | 48.9 | 55.3 | 54.3 | 58.9 | 54.8 | 53.6 | 46.2 | 69.7 | 76.2 | 66.2 | 60.9 | 69.6 | 66.6 | **58.1** |

