# OpenReview forum: "Towards Stable Test-time Adaptation in Dynamic Wild World"
_ICLR.cc/2023/Conference — ICLR 2023 notable top 5%_

### Official Review · Reviewer_Ww1T · 2022-10-20

**Confidence:** 4
**Correctness:** 4
**Technical Novelty And Significance:** 3
**Empirical Novelty And Significance:** 4
**Recommendation:** 8

**Clarity, Quality, Novelty And Reproducibility:**

Overall the paper produces the settings, approaches, and experiments clearly and it is easy to follow the paper. The main novelty of the paper lies in the "wild" setting; the proposed procedure SAR is composed of well known components and not highly novel but still a useful contribution for approached the novel "wild" setting of TTA. The quality of the empirical evaluation is good even though a few improvements/extensions could be made (see above).
Reproducibility is given as details of the methods and experiments have been discussed and source code will be released upon acceptance. Moreover, experiments have been conducted based upon publicly available model checkpoints.

Minor:
 * it was unclear to me which parameters have been adapted: only the affine parameters of normalization layers or all network parameters?

**Strength And Weaknesses:**

Strengths:
 * the proposed setting of "wild" test-time adaptation is of high practical relevancy and it is considerably more challenging that the standard setting of batched, single-target domain, class-balanced TTA. The authors also motivate and illustrate this setting well (compare Figure 1)
 * the paper does a good job in empirically identifying problematic parts of existing TTA approaches in the "wild" setting, namely (i) batch-norm's implicit single-domain assumption and reliance on large batches and (ii) model collapse caused by large gradient norms.
 * Pragmatic solutions to these issues are identified that are not inherently novel but shown to be very efficient in the "wild" setting: (i) using group/layer norm instead of batch-norm and (ii) using SAR (SAM + entropy-based filtering + model recovery) to prevent model collapse.
 * Overall, the paper is strong on the empirical side and provides an extensive set of experiments on the different parts of the "wild" setting and the different components of the proposed wild TTA procedure.
 * It is laudable that runtimes of different TTA-procedures are compared in Table 1. I would recommend to use the same unit (seconds) for all methods though. One question: why has SAR nearly the same runtime as TENT? I would expect that the double backward pass causes it to be somewhat slower.

Weaknesses:
 * empirical evaluation is limited to target domains with common corruptions (e.g. ImageNet-C); investigation of other types of domain shifts like ImageNet-R would provide additional insights in terms of the generality of the proposed procedures.
 * the paper's comparison of normalization layers has a strong confounding factor in the overall model architecture: while batch-norm and group-norm are studied as part of a ResNet-50, layer-norm is studied as part of an ViT. Thus, a statement like "GN models are more suitable for self-supervised TTT than LN models since TTT+VitBase-LN is sensitive to different sample orders and has large variances over different runs." is problematic since it might not be that differences are due to the normalization layer (but rather to the model architectures). It would be preferable to remove the confounding factor of model architecture in this part.
 * The ablation study of SAR in Table 5 is nice but should be extended to cover all components of the wild setting and not only imbalanced labels
 * the model recovery scheme seems to be very helpful in the ViT-LN setting. To which extent would it also benefit TENT and EATA? That is: to which extent are benefits of SAR over TENT/EATA due to model recovery and not due to SAR?
 * not really a weakness, but a natural question arising would be: how would a model trained with SAM on the source domain perform in the comparisons given that SAM was found to be helpful at test-time?


**Summary Of The Paper:**

The paper focuses on fully test-time adaptation in dynamic and "wild" settings, in which (i) data from different target domains might be mixed at test time, (ii) test samples might not come in batches, and (iii) test samples might not be independently sampled in the sense that class frequency p(y) differs across batched. The authors identify batch norm as being harmful under such a setting and identify networks using group or layer norm to be preferable. Moreover, the authors propose sharpness-aware and reliable entropy minimization (SAR), to prevent model collapse caused by samples with large gradient norms. SAR consists of using the sharpness aware minimizer (SAM), entropy-based filtering of test samples used for adaptation, and a heuristic model recovery scheme. An empirical study in the dynamic "wild" settings and the proposed approaches is conducted on domain shifts with common corruptions.


**Summary Of The Review:**

Overall, I think this paper is a solid contribution that deserves acceptance at ICLR. The main contribution for me is to study and extend TTA in a more realistic "wild" setting, which should increase practical relevancy of TTA. The methods themselves are not novel but their combination and fit to "wild" TTA are sufficiently novel.

---

> ### Author Response · Authors · 2022-11-13
> **Responses to Reviewer Ww1T [1/3]**
>
> We deeply appreciate your valuable feedback and constructive comments on improving our paper. We would like to address your questions below.
>
> >Q1. It is laudable that runtimes of different TTA-procedures are compared in Table 1. I would recommend to use the same unit (seconds) for all methods though. One question: why has nearly the same runtime as TENT? I would expect that the double backward pass causes it to be somewhat slower.
>
> A1. Thank you for your valuable suggestion. We have revised Table 1 and used the same unit (seconds). Our SAR has nearly the same runtime as Tent because it removes many samples from adaptation and reduces the total number of forward and backward passes. Table A summarizes the number of forward and backward passes of Tent and SAR. More comparisons of this metric can be found in Table 6 in Appendix C.2.
>
> Table A. The number of forward and backward propagation of Tent and our SAR with ResNet50-GN on ImageNet-C (Gaussian noise, severity level 5).
> |            |    #Forward     | #Backward |  GPU time |
> | ---------- |:---------------:|:---------:|:---------:|
> | Tent       |     50,000      |  50,000   |110 seconds|
> | SAR (ours) | 50,000+12,710   | 12,710×2  |115 seconds|
>
>
> > Q2. Empirical evaluation is limited to target domains with common corruptions (e.g. ImageNet-C); investigation of other types of domain shifts like ImageNet-R would provide additional insights in terms of the generality of the proposed procedures.
>
> A2. Thank you for your valuable suggestion. We conduct experiments on ImageNet-R under two wild settings (label shifts and batch size 1) in Table B. The overall results are consistent with that on ImageNet-C: 1) ResNet50-GN and VitBase-LN perform more stable than ResNet50-BN; 2) Compared with Tent and EATA, our SAR achieves the best performance on ResNet50-GN and VitBase-LN. We have included these results in the revision.
>
> Table B. Comparison with online test-time adaptation methods on ImageNet-R. We report accuracy (%) under two wild settings, i.e., **online imbalanced label distribution shifts** and **single sample adaptation (batch size = 1)**.
> |            | Label shifts | Label shifts | Label shifts |   BS=1   |   BS=1   |    BS=1    |
> |:---------- |:------------:|:------------:|:------------:|:--------:|:--------:|:----------:|
> | Method     |   Res50-BN   |   Res50-GN   |  VitBase-LN  | Res50-BN | Res50-GN | VitBase-LN |
> | No adapt   |     36.2     |     40.8     |     43.1     |   36.2   |   40.8   |    43.1    |
> | Tent       |     6.6      |     41.7     |     45.2     |   0.6    |   42.2   |    40.5    |
> | EATA       |     5.8      |     40.9     |     47.5     |   0.6    |   42.3   |    52.5    |
> | SAR (ours) |      -       |   **42.9**   |   **52.0**   |    -     | **43.9** |  **53.1**  |
>
>
>
> > Q3. The paper's comparison of normalization layers has a confounding factor in the overall model architecture: while batch-norm and group-norm are studied as part of a ResNet-50, layer-norm is studied as part of an ViT. Thus, a statement like "GN models are more suitable for self-supervised TTT than LN models since ..." is problematic. It would be preferable to remove the confounding factor of model architecture in this part.
>
>
>
> A3. Thank you for pointing out this. We choose ResNet50-BN, ResNet50-GN and VitBase-LN to investigate the norm layer effects since for mainstream neural networks the norm layers are often coupled with network architecture. Specifically, GN and BN are often combined with conventional networks, while LN is more suitable for transformer networks. We have revised our statements from the experiments in Section 4 by considering the confounding architecture factor. For example, a statement like "ResNet50-GN is more suitable for self-supervised TTT than VitBase-LN since TTT+VitBase-LN is sensitive to different sample orders and has large variances over different runs".
>
> We also conduct more experiments to compare the performance of online entropy minimization on ResNet50-BN (as used in the main paper) and ConvNeXt-LN [1] (from timm repository). ConvNeXt is a convolutional network equipped with LN. The authors conduct significant modifications over ResNet to make this LN-based Conv network work well, such as modifying the architecture (ResNet block to ConvNeXt block, activation functions, etc.), various training strategies (stochastic depth, random erasing, EMA, etc.). From Table C, Tent+ConvNeXt-LN performs more stable than Tent+ResNet50-BN, but still suffers several failure cases. These results are consistent with that of ResNet50-BN vs. VitBase-LN. We have included these results in the revision.

---

> > ### Author Response · Authors · 2022-11-13
> > **Responses to Reviewer Ww1T [2/3]**
> >
> > Table C. Results of Tent on ResNet50-BN and ConvNeXt-LN. We report **accuracy (%)** on ImageNet-C **under online imbalanced label distribution shifts**.
> > | Severity | Model+Method | Gauss. | Shot | Impul. | Defoc. | Glass | Motion | Zoom | Snow | Frost | Fog  | Brit. | Contr. | Elastic | Pixel | JPEG | Avg. |
> > |:--:| :--- |:---:|:---:|:--:|:------:|:-----:|:------:|:----:|:----:|:-----:|:----:|:-----:|:------:|:-------:|:-----:|:----:|:----:|
> > | level 5  | ResNet50-BN  | 2.2    | 2.9  | 1.8    | 17.8   | 9.8   | 14.5   | 22.5 | 16.8 | 23.4  | 24.6 | 59.0  | 5.5    | 17.1    | 20.7  | 31.6 | 18.0 |
> > | level 5  | +Tent        | 1.2    | 1.4  | 1.4    | 1.0    | 0.9   | 1.2    | 2.6  | 1.7  | 1.8   | 3.6  | 5.0   | 0.5    | 2.6     | 3.2   | 3.1  | 2.1  |
> > | level 5  | ConvNeXt-LN  | 52.3   | 52.7 | 52.3   | 31.7   | 18.7  | 42.5   | 38.1 | 54.2 | 58.3  | 50.6 | 75.6  | 56.8   | 32.2    | 39.2  | 60.4 | 47.7 |
> > | level 5  | +Tent        | 26.3   | 11.9 | 36.9   | 31.1   | 12.7  | 14.6   | 5.1  | 7.8  | 5.3   | 6.6  | 79.0  | 67.6   | 1.5     | 68.4  | 65.8 | 29.4 |
> > |          |              |        |      |        |        |       |        |      |      |       |      |       |        |         |       |      |      |
> > | level 3  | ResNet50-BN  | 27.7   | 25.2 | 25.1   | 37.8   | 16.7  | 37.8   | 35.3 | 35.2 | 32.1  | 46.7 | 69.5  | 46.2   | 55.4    | 46.2  | 59.4 | 39.8 |
> > | level 3  | +Tent        | 3.4    | 3.2  | 3.2    | 2.3    | 2.0   | 2.4    | 3.4  | 2.4  | 2.4   | 4.6  | 5.4   | 3.0    | 4.8     | 4.6   | 4.5  | 3.4  |
> > | level 3  | ConvNeXt-LN  | 71.1   | 69.8 | 72.4   | 55.2   | 36.6  | 64.3   | 52.8 | 63.2 | 64.0  | 66.0 | 79.4  | 75.8   | 69.1    | 68.5  | 71.9 | 65.4 |
> > | level 3  | +Tent        | 73.4   | 73.5 | 74.4   | 66.8   | 61.6  | 71.4   | 64.2 | 22.6 | 39.5  | 76.8 | 81.2  | 78.9   | 76.8    | 75.6  | 75.8 | 67.5 |
> >
> > [1] A ConvNet for the 2020s. CVPR, 2022.
> >
> > > Q4. The ablation study of SAR in Table 5 is nice but should be extended to cover all components of the wild setting and not only imbalanced labels
> >
> > A4. Thank you for your valuable suggestion. We have conducted more ablation experiments under the settings of "batch size (BS)=1" in Table D and "mix domain shifts" in Table E. The results are generally consistent with that in Table 5. Both the reliable entropy and sharpness-aware optimization work together to stabilize online TTA. It is worth noting that only in VitBase-LN under BS=1 the model recovery scheme is activated and improves the average accuracy from 55.7% to 56.4%. We have included these results in the revision.
> >
> > Table D. Effects of components in our SAR. We report the **accuracy (%)** **under batch size=1** on ImageNet-C of severity level 5.
> > | Model+Method          | Avg. | Gauss. | Shot | Impul. | Defoc. | Glass | Motion | Zoom | Snow | Frost | Fog  | Brit. | Contr. | Elastic | Pixel | JPEG |
> > | --------------------- |:----:|:------:|:----:|:------:|:---:|:-----:|:------:|:----:|:----:|:-----:|:----:|:-----:|:------:|:-------:|:-----:|:----:|
> > | ResNet50 (GN)+Entropy | 24.7 |  3.4   | 4.5  |  4.0   |  16.7  |  5.9  |  27.0  | 29.7 | 16.3 | 26.6  | 2.1  | 72.1  |  46.4  |   7.5   | 52.5  | 56.2 |
> > | +reliable             | 32.8 |  22.8  | 25.5 |  23.0  |  18.4  | 14.8  |  27.0  | 28.6 | 40.0 | 43.3  | 18.4 | 71.5  |  43.1  |  15.2   | 45.6  | 55.4 |
> > | +reliable+sa           | 34.4 |  23.8  | 26.4 |  24.0  |  18.6  | 15.4  |  28.3  | 30.5 | 44.8 | 44.8  | 26.7 | 72.4  |  44.5  |  12.2   | 46.9  | 56.1 |
> > | +reliable+sa+recover  | 34.2 |  23.8  | 26.4 |  23.9  |  18.5  | 15.4  |  28.3  | 30.6 | 44.6 | 44.9  | 24.7 | 72.4  |  44.4  |  12.3   | 47.0  | 56.1 |
> > |                       |      |        |      |        |        |       |        |      |      |       |      |       |        |         |       |      |
> > | VitBase (LN)+Entropy  | 48.8 |  42.9  | 1.4  |  43.9  |  52.6  | 48.8  |  55.8  | 51.4 | 22.2 | 20.1  | 67.0 | 75.1  |  64.7  |  53.7   | 67.2  | 64.5 |
> > | +reliable             | 50.5 |  34.8  | 4.2  |  35.5  |  50.5  | 45.9  |  54.0  | 48.6 | 52.5 | 47.8  | 65.5 | 74.5  |  63.4  |  51.4   | 65.6  | 63.0 |
> > | +reliable+sa           | 55.7 |  40.4  | 37.3 |  41.2  |  53.6  | 50.6  |  57.3  | 53.0 | 58.7 | 40.9  | 68.8 | 75.9  |  65.7  |  57.9   | 69.0  | 66.0 |
> > | +reliable+sa+recover  | 56.4 |  40.4  | 37.3 |  41.2  |  53.6  | 50.6  |  57.3  | 53.0 | 58.7 | 50.7  | 68.8 | 75.4  |  65.7  |  57.9   | 69.0  | 66.0 |
> >
> > Table E. Effects of components in our SAR. We report the accuracy **under mixed domain shifts**, i.e., mixture of 15 corruption types of ImageNet-C with severity level 5.
> > | Model+Method          | Accuracy (%) |
> > |:------- |:--------:|
> > | ResNet50 (GN)+Entropy |     33.5     |
> > | +reliable             |     38.1     |
> > | +reliable+sa          |     38.2     |
> > | +reliable+sa+recover  |     38.2     |
> > |||
> > | VitBase (LN)+Entropy  |     18.5     |
> > | +reliable             |     55.2     |
> > | +reliable+sa          |     57.2     |
> > | +reliable+sa+recover  |     57.2     |

---

> > > ### Author Response · Authors · 2022-11-13
> > > **Responses to Reviewer Ww1T [3/3]**
> > >
> > > >Q5. The model recovery scheme seems to be very helpful in the ViT-LN setting. To which extent would it also benefit TENT and EATA? That is: to which extent are benefits of SAR over TENT/EATA due to model recovery and not due to SAR?
> > >
> > > A5. Thank you for your valuable question. We further apply our Model Recovery scheme to Tent and EATA. From Table F, the model recovery indeed helps Tent a lot (e.g., the avg accuracy from 22.0% to 26.1% on ResNet50-GN) while its performance gain on EATA is a bit marginal. Compared with Tent+recovery and EATA+recovery, our SAR greatly boosts the adaptation performance, e.g., the average accuracy 26.1% (Tent+recovery) vs. 37.2% (SAR) on ResNet50-GN, suggesting the effectiveness of the proposed SAR.
> > >
> > > Table F. Results of combining model recovery scheme with Tent and EATA. We report the **accuracy (%)** on ImageNet-C severity level 5 **under online imbalanced label distribution shifts**.
> > > | Model+Method | Avg. | Gauss. | Shot | Impul. | Defoc. | Glass | Motion | Zoom | Snow | Frost | Fog | Brit. | Contr. | Elastic | Pixel | JPEG |
> > > | :- |:-:|:-:|:-:|:-:|:-:|:-:|:-:|:-:|:-:|:-:|:-:|:-:|:-:|:-:|:-:|:-:|
> > > | ResNet50 (GN) | 30.6 |18.0|19.8|17.9|19.8|11.4|21.4|24.9|40.4|47.3|33.6|69.3|36.3|18.6|28.4|52.3|
> > > | +Tent | 22.0 | 2.6 | 3.3 | 2.7 | 13.9 | 7.9 | 19.5 | 17.0 | 16.5 | 21.9 | 1.8 | 70.5 | 42.2 | 6.6 | 49.4 | 53.7 |
> > > | +Tent+recover | 26.1 | 10.1 | 12.2 | 10.6 | 13.9 | 8.5 | 19.5 | 20.6 | 24.3 | 33.5 | 8.9 | 70.5 | 42.2 | 13.5 | 49.4 | 53.7 |
> > > | +EATA | 31.6 | 27.0 | 28.3 | 28.1 | 14.9 | 17.1 | 24.4 | 25.3 | 32.2 | 32.0 | 39.8 | 66.7 | 33.6 | 24.5 | 41.9 | 38.4 |
> > > | +EATA+recover | 32.9 | 26.1 | 31.0 | 27.2 | 19.9 | 18.5 | 25.7 | 25.7 | 35.9 | 28.6 | 40.4 | 68.2 | 35.3 | 27.6 | 42.9 | 40.9 |
> > > | +SAR (ours) | **37.2** | 33.1 | 36.5 | 35.5 | 19.2 | 19.5 | 33.3 | 27.7 | 23.9 | 45.3 | 50.1 | 71.9 | 46.7 | 7.1 | 52.1 | 56.3 |
> > > ||||||||||||||||||
> > > | VitBase (LN) | 29.9 | 9.4 | 6.7 | 8.3 | 29.1 | 23.4 | 34.0 | 27.0 | 15.8 | 26.3 | 47.4 | 54.7 | 43.9 | 30.5 | 44.5 | 47.6 |
> > > | +Tent | 47.3 | 32.7 | 1.4 | 34.6 | 54.4 | 52.3 | 58.2 | 52.2 | 7.7 | 12.0 | 69.3 | 76.1 | 66.1 | 56.7 | 69.4 | 66.4 |
> > > | +Tent+recover | 52.3 | 40.3 | 10.1 | 42.4 | 54.4 | 52.3 | 58.1 | 52.2 | 31.6 | 39.2 | 69.3 | 76.1 | 66.1 | 56.7 | 69.4 | 66.4 |
> > > | +EATA | 50.0 | 35.9 | 34.6 | 36.7 | 45.3 | 47.2 | 49.3 | 47.7 | 56.5 | 55.4 | 62.2 | 72.2 | 21.7 | 56.2 | 64.7 | 63.7 |
> > > | +EATA+recover | 49.9 | 35.9 | 34.6 | 36.7 | 45.3 | 47.2 | 49.3 | 47.7 | 56.5 | 55.4 | 62.2 | 72.2 | 21.7 | 56.2 | 64.7 | 63.7 |
> > > | +SAR (ours) | **58.1** | 46.5 | 43.1 | 48.9 | 55.3 | 54.3 | 58.9 | 54.8 | 53.6 | 46.2 | 69.7 | 76.2 | 66.2 | 60.9 | 69.6 | 66.6 |
> > >
> > >
> > > > Q6. Not really a weakness, but a natural question arising would be: how would a model trained with SAM on the source domain perform in the comparisons given that SAM was found to be helpful at test-time?
> > >
> > > A6. Training a model on the source domain via SAM also boosts the model accuracy on both clean ImageNet validation set and ImageNet-C (as shown in Table G). A recent work [1] thoroughly investigates the generalization ability of models trained on the source domain via SAM. Here, we further apply our SAR to this VitBase-SAM model. Results in Table H show that SAR further boosts the adaptation performance, suggesting its effectiveness.
> > >
> > > Table G. Accuracy (%) of VitBase that trained on source domain with and without SAM. **Results are directly copied from [1]**. The **ImageNet-C accuracy is measured against 19 corruptions with all 5 levels [1]**. Note that here 1) the evaluation involves no test-time adaptation, i.e., the model parameters are frozen during testing; 2) the source domain only consists of training data from ImageNet-1k, with no additional broad pre-training data such as ImageNet-21k.
> > > | | VitBase | VitBase-SAM |
> > > | ---------- |:-------:|:-----------:|
> > > | ImageNet | 74.6 | **79.9** |
> > > | ImageNet-C | 46.6 | **56.5** |
> > >
> > >
> > > Table H. Performance of SAR on VitBase-SAM. We report the **average accuracy (%) over 15 corruption types** on ImageNet-C under **online imbalanced label distribution shifts.**
> > > | Severity | Model+Method | Average accuracy |
> > > |:- |:-:|:-:|
> > > | Level 5 | VitBase-SAM | 24.6 |
> > > | Level 5 | +SAR (ours) | **41.8** |
> > > | Level 3 | VitBase-SAM | 49.6 |
> > > | Level 3 | +SAR (ours) | **65.1** |
> > >
> > > [1] When vision transformers outperform ResNets without pre-training or strong data augmentations. ICLR, 2022.
> > >
> > > > Q7. It was unclear to me which parameters have been adapted: only the affine parameters of normalization layers or all network parameters?
> > >
> > > A7. We update only affine parameters of normalization layers (as described in the second paragraph of Section 5). We have made this clearer in the revision.
> > >
> > > We thank you for appreciating our contributions. We sincerely hope our clarifications above have addressed your questions.

---

> > > > ### Comment · Reviewer_Ww1T · 2022-11-16
> > > > **Feedback to the response**
> > > >
> > > > I would like to thank the authors for their extensive response. All relevant discussion items from my review have been addressed satisfyingly. I am keeping my overall recommendation "8: accept, good paper" but increase the score for Correctness to "4: All of the claims and statements are well-supported and correct.".

---

> > > > > ### Author Response · Authors · 2022-11-16
> > > > > **Thank you for appreciating our response!**
> > > > >
> > > > > Dear Reviewer,
> > > > >
> > > > > We are glad to know that our response has addressed your questions. We would like to thank you again for appreciating our work and recognizing our contributions!
> > > > >
> > > > > Best,
> > > > >
> > > > > The Authors

---

### Official Review · Reviewer_w35R · 2022-10-24

**Confidence:** 4
**Correctness:** 3
**Technical Novelty And Significance:** 3
**Empirical Novelty And Significance:** 3
**Recommendation:** 8

**Clarity, Quality, Novelty And Reproducibility:**

This paper is easy to read, novel, and of high quality. I think it has a big chance of being reproduced.

**Strength And Weaknesses:**

Strength:
1. The authors introduce wild online TTA settings. For example, the test data may have 1) mixture distribution variation, 2) mini-batch, 3) online imbalanced label distribution variation, which are realistic during application.

2. Extensive experiments are conducted to illustrate the limitations of Batch Norm in TTA and further observed that batch-independent norms (e.g., Group Norm and Layer Norm) will be more stable than Batch Norm. The experiments provide new insights for the TTA community.

3. The authors observe that models with Group Norm or Layer Norm tend to collapse, especially when the severity of the distribution shift is high. They then propose a sharpness-aware learning scheme to alleviate the collapse problem, which is technically reasonable and novel within the scope of TTA.

4. The authors propose a sharpness-aware and reliable entropy minimization method to uniformly address the instability of previous methods under three wild TTA settings. Extensive experiments on ImageNet-C demonstrate the effectiveness of SAR.

Weakness:
1. This paper focuses on dynamic wild-test time adaptation settings. I am curious about under what circumstances does the third situation arise? Can the author provide some examples?

2. In Tables 2 and 4, all considered methods (including the proposed method) achieve worse TTA performance than baselines on ResNet50-GN with damaged Defoc. That is, all SOTA TTA methods cannot fit out-of-distribution samples. Could the authors discuss more about this phenomenon?

3. In Section 4, the description is a bit difficult to understand. Since these figures are informative, it is best to indicate which subgraph is being described when describing.

4. The numbers in section 4 are somewhat difficult to understand directly and the font size is too small. It's best to optimize these numbers for readability.

5. In Table 2, were the results for different damages measured with the same sample sequence?




**Summary Of The Paper:**

This paper proposes a new method named sharpness-aware and reliable entropy minimization (SAR) to stabilize wild TTA. Specifically, to mitigate the collapse of group norm or layer norm models, SAR first filters samples with high and noisy gradients according to entropy loss. Furthermore, SAR introduces a sharpness-aware learning scheme to ensure that model weights remain at a flat minimum during optimization. Extensive experiments demonstrate the importance of Wild TTA and the effectiveness of SAR.

**Summary Of The Review:**

This paper is easy to read and novel but still has some problems. Please refer to the weakness.

---

> ### Author Response · Authors · 2022-11-13
> **Responses to Reviewer w35R**
>
>
> We deeply appreciate your positive feedback and constructive comments on improving our paper. We will address your questions below.
>
>
> > Q1. This paper focuses on dynamic wild-test time adaptation settings. I am curious about under what circumstances does the third situation arise? Can the author provide some examples?
>
> A1. Thank you for your valuable question. The third situation in Figure 1 \(c\) means that the label distribution $Q(y)$ of test samples may online change and be imbalanced at different time steps. This situation is common in practice. For instance, in traffic surveillance scenarios, the camera may first capture a sequence of test images with the class label "sedan" (at $t=1$) and then a sequence of test images with the class label "truck" (at $t=2$). In this case, the corresponding label distributions are $Q_1(y)=[p_{sedan}=1,0,0,0,...]$ and $Q_2(y)=[0,p_{truck}=1,0,0,...]$.
>
>
>
>
> > Q2. In Tables 2 and 4, all considered methods (including the proposed method) achieve worse TTA performance than baselines on ResNet50-GN with damaged Defoc. That is, all SOTA TTA methods cannot fit out-of-distribution samples. Could the authors discuss more about this phenomenon?
>
>
> A2. The performance of TTA methods is affected by both the nature of the test samples and the initial model weights. For the Defocus corruption, TTA methods actually achieve better accuracy than No Adapt on VitBase(LN)-Level5 (in Tables 3-4) and ResNet50(GN)-Level3 (in Tables 8-9), while on ResNet50(GN)-Level5 all TTA methods have inferior accuracy than No Adapt. This indicates that the (current) compared TTA methods may not be very suitable for the adopted ResNet50(GN) model on Defocus with level 5. Exploring which model weights and data are suitable for performing TTA is also an interesting and important direction, which we leave to the future.
>
> Moreover, it is worth noting that although all compared TTA methods perform worse than No Adapt on Defocus with ResNet50-GN, our SAR still performs the best and suffers the smallest performance degradation (e.g., SAR degrades 0.5% while Tent degrades 5.8%), suggesting the effectiveness of SAR.
>
>
>
>
> > Q3. In Section 4, the description is a bit difficult to understand. Since these figures are informative, it is best to indicate which subgraph is being described when describing.
>
> A3. Thank you for your valuable suggestion. We have clearly referred to the subfigures when describing them in the revised paper.
>
>
> > Q4. The numbers in Section 4 are somewhat difficult to understand directly and the font size is too small. It's best to optimize these numbers for readability.
>
> A4. Thank you for your valuable suggestion. We have revised the numbers and the font size in the figures in Section 4 to make them more readable.
>
>
> > Q5. In Table 2, were the results for different damages measured with the same sample sequence?
>
> A5. Yes, we measure different methods and damages with the same sample sequence for fair comparisons. We have made this clearer in the revision.
>
> We thank you for appreciating our contributions. We sincerely hope our clarifications above have addressed your questions.

---

> ### Author Response · Authors · 2022-11-17
> **Thanks to Reviewer w35R**
>
> Dear Reviewer w35R,
>
> We would like to thank you again for recognizing the novelty and contributions of our work and also for the valuable feedback. We are wondering if our response has addressed your questions appropriately.
>
> Kindly let us know if your might have further comments, and we will do our best to address them.
>
> Best,
>
> The Authors

---

### Official Review · Reviewer_AHGC · 2022-10-26

**Confidence:** 4
**Correctness:** 4
**Technical Novelty And Significance:** 4
**Empirical Novelty And Significance:** 3
**Recommendation:** 8

**Clarity, Quality, Novelty And Reproducibility:**

This paper is clearly presented in high quality. Their method is quite novel. Necessary implementation details required to reproduce their results are provided.

**Strength And Weaknesses:**

* Strengths:

    1. This work provides detailed investigation of unstable reasons and careful analysis of failure cases in TTA. According to their investigation and analysis, a sharpness-aware and reliable method is proposed to solve those issues.

    2. Their discussion about norm layer is interesting. They also find weaknesses of only using GN and LN and provide a good solution to deal with such issues. The analysis of test sample gradients and the corresponding removing method is quite novel and efficient.

    3. Their experiments and ablation studies demonstrate that their analysis and hypothesis is indeed true, and the method they proposed is effective to improve the performance of TTA on ImageNet-C.

* Weaknesses:

    1. Their analysis about the norm layer is mainly based on empirical evidences. It is good to support their study with comprehensive experiments. However, I still wonder is there any intuitive explanation about choosing GN and LN instead of BN? This could help readers to better understand the analysis.

    2. In Fig. 6, the authors find even with fine-tuned hyper-parameters, gradient clipping yield worse results than no clipping, which is a bit counter-intuitive. Is there any explanation about the failure of gradient clipping even with fine-tuning?

**Summary Of The Paper:**

The authors propose a test-time adaptation (TTA) method to deal with distribution shifts between training and test data. Their method aims to deal with the unstable issue of online model updating in TTA. A sharpness-aware and reliable entropy minimization method is designed. They also remove partial noisy samples and encourage model weights to go to a flat minimum to further stabilize TTA.

**Summary Of The Review:**

Please address concerns mentioned in weaknesses.

---

> ### Author Response · Authors · 2022-11-13
> **Responses to Reviewer AHGC**
>
> We thank you for recognizing the novelty and contributions of our paper and also for the positive feedback. We would like to answer your questions below.
>
>
>
> > Q1. Their analysis about the norm layer is mainly based on empirical evidences. It is good to support their study with comprehensive experiments. However, I still wonder is there any intuitive explanation about choosing GN and LN instead of BN? This could help readers to better understand the analysis.
>
> A1. Thank you for your valuable question. Prior TTA methods are often built upon BN statistics adaptation, which exploits test data to calculate the mean and variance in BN layers. However, under three wild test settings (in Figure 1), the mean and variance estimation will be problematic:
>
> 1. For **mixed domain shifts:** BN statistics actually represent a distribution and ideally each distribution should have its own statistics. Simply estimating shared BN statistics of multiple distributions from mini-batch test samples unavoidably obtains limited performance.
> 2. For **small test batch size:** the quality of estimated statistics relies on the batch size, and it is hard to use very few samples to estimate it accurately.
> 3. For **online imbalanced label distribution shifts:** the imbalanced label shift will also result in biased BN statistics towards some specific classes in the dataset.
>
> However, in batch-agnostic (agnostic to the way samples are grouped into a batch) norm layers like GN and LN, the statistics are calculated over every single sample across different channels and thus do not suffer this problematic statistics estimation issue. We also mentioned these intuitive explanations in the second paragraph of Section 3.1.
>
>
>
> > Q2. In Fig. 6, the authors find even with fine-tuned hyper-parameters, gradient clipping yield worse results than no clipping, which is a bit counter-intuitive. Is there any explanation about the failure of gradient clipping even with fine-tuning?
>
> A2. Thank you for your valuable question. Actually, in Fig. 6, all variants (clip by value and by norm) of **gradient clipping** with different hyper-parameters **outperform "no clip" regarding the overall accuracy at the last batch**. Moreover, with carefully fine-tuned hyper-parameters $\delta$, gradient clipping also achieves better performance than "no adapt". To be specific,
>
>  - In the left of Fig. 6, clip by value with "max_value=0.001" outperforms  "no adapt";
>  - In the right of Fig. 6, clip by norm with "max_norm=0.1" outperforms "no adapt".
>
> Nonetheless, the performance gain of gradient clipping over "no adapt" is very marginal, since this $\delta$ is actually very small and thus would limit the learning ability of the model. Besides, the clipped gradients may also point in a very different direction from the true gradients. Meanwhile, if we set a larger value of $\delta$, gradient clipping may fail to stabilize the adaptation process and the accuracy will degrade after the model collapses (worse than "no adapt"). We have made this clearer in the revision.
>
> We thank you for appreciating our contributions. We sincerely hope our clarifications above have addressed your questions.

---

> ### Author Response · Authors · 2022-11-17
> **Thanks to Reviewer AHGC**
>
> Dear Reviewer AHGC,
>
> We would like to thank you again for recognizing the novelty and contributions of our work and also for the valuable feedback. We are wondering if our response has addressed your questions appropriately.
>
> Kindly let us know if your might have further comments, and we will do our best to address them.
>
> Best,
>
> The Authors

---

### Official Review · Reviewer_7Sq7 · 2022-10-26

**Confidence:** 3
**Clarity, Quality, Novelty And Reproducibility:** The proposed method does not strike m…
**Correctness:** 3
**Technical Novelty And Significance:** 2
**Empirical Novelty And Significance:** 2
**Recommendation:** 5

**Strength And Weaknesses:**

**Strengh**
The paper is easy to follow. Authors also include explanations/intuitions to certain phenomena and design choices.

**Weakness:** Follows are my concerns. More detail below.
- The proposed method somewhat lacks novelty.
- The empirical evaluation can be improved and strengthened.

**Summary Of The Paper:**

The paper investigates the pitfalls of online domain adaptation, that the model tends to collapse with small batch sizes and class imbalances at test time. The authors also propose two techniques to mitigate the problems.

**Summary Of The Review:**

Below are my concerns, which I hope the authors will address:
**The two solutions proposed by the authors do not seem new to me**:
- The entropy selective term $S(x)$ looks a lot like the one from EATA. In fact, it is almost the same thing, just without the coefficient term. It is also worrying to me that the authors do not mention nor discuss this similarity.
- The sharpness-aware minimization technique is also not new. It is adopted directly from SAM from the supervised loss to the self-supervised loss. On a side note, the authors mentioned a problem with gradient selection/clipping is that the threshold needs careful tuning. However, I would also argue that the hyper-parameter $\rho$ for the sharpness-aware entropy minimization also needs careful tuning, since it should depend on the norm of the parameters of the source-pretrained model.

**The empirical evaluation seems incomprehensive**
- The authors only validate their method for the case of entropy minimization. To the best of my knowledge, there are more surrogate losses that are often used for ODA. It would be beneficial to test if the method also helps in these cases.
- I wonder why the authors do not test their method with BN models (e.g., original resnet50)? If it does not perform well in this case, I suggest adding some explanation/discussion regarding this.
- It would also be beneficial to conduct experiments on domain shift datasets such as VisDA.


===========================
Post rebuttal:

I would like to thank the authors for their detailed response and the effort they put into this rebuttal.

They addressed most of my concerns, but some points remained (detailed below). For this reason, I increase my score from 3 to 5, since I think some arguments still need to be sharpened in the final draft.

- In the comparison with EATA and SAM, the authors pointed out some differences between their proposed method and these existing works (mostly about the purposes of the work, not much about the methodologies). With such similarities in the methodologies of the methods, it is absolutely important to acknowledge and discuss these details in the paper. As far as I am aware, the authors did not include any new discussions regarding this in the revision. Note that it is okay to (borrow)/(take motivation from) existing works, but discussion about the similarities, differences and novelties are needed for completeness.

- The authors mentioned they do not apply their method to other test time adaptation method, such as TTT, TTT+,... since they are computationally expensive (require several forward/backward passes). I find this arguement not convincing. If some methods are more expensive but offer better performance, they are still worth some value (some people even prefer these methods more for the performance). As the paper is deadling with TTA in the wild, I think it is important to cover a wide range of settings, including the TTA method.

---

> ### Author Response · Authors · 2022-11-13
> **Responses to Reviewer 7Sq7 [1/3]**
>
> Thank you for taking the time to review our paper and providing valuable feedback. We would like to answer your questions below.
>
> > Q1. The proposed SAR method somewhat lacks novelty.
>
>
> A1. In this paper, we address a very practical and challenging problem, namely online test-time adaptation (TTA) under wild test settings. To our best knowledge, we are the first   to investigate and analyze why prior online TTA methods fail to perform stably under wild scenarios. We find that 1) BN is a crucial obstacle hindering online TTA's stability and 2) noisy/large gradients would cause model collapse during online TTA. We thus propose an efficient sharpness-aware and reliable (SAR) entropy minimization method to enable online TTA to be stable under wild test settings.
>
>
> We would like to highlight our **novelty and contributions** in the following three folds:
> 1. **TTA under novel "wild test settings":**
> - We study and extend TTA in realistic "wild test settings" (mix domain shifts, single sample, online imbalanced label distribution shifts) to increase the practical relevancy of TTA. These wild scenarios are common in real-world applications and more challenging than prior standard settings of single domain shift, batched, class-balanced TTA. In this paper, we analyze the failure reasons of prior methods under wild test settings and then propose corresponding solutions.
> - This novelty is highly recognized by other reviewers:
>     * *"The authors introduce wild online TTA settings. ..., which are **realistic during application**"* (by Reviewer w35R)
>     * *"The main novelty of the paper lies in the "wild" setting. The proposed setting of "wild" test-time adaptation is of **high practical relevancy** and it is considerably **more challenging**"* (by Reviewer Ww1T)
> 2. **Investigations about reasons that lead to unstable online TTA**:
> - We identify that under wild settings the mean and variance estimation in BN layers would be problematic and thus harm the online model adaptation, resulting in unstable TTA performance (see Sections 3.1 and 4).
> - We identify that some noisy/large gradients of entropy loss would cause the model to be optimized to collapsed trivial solutions, i.e., assigning the same class label for all samples (see Section 3.1).
> - This novelty is highly recognized by other reviewers:
>     * *"... **provide new insights for the TTA community**"* (by Reviewer w35R)
>     * *"Their **discussion** about norm layer is **interesting**. The analysis of test sample gradients and the corresponding removing method is **quite novel and efficient**"* (by Reviewer AHGC)
> 3. **New stable TTA method**:
> - We propose a SAR method to conquer the effects of noisy/large gradients and thus address the model collapse issue. SAR properly introduces 1) selective entropy minimization, 2) sharpness-aware optimization and 3) a model recovery scheme **to make online TTA stabilized under wild test settings**.
> - This novelty is highly recognized by other reviewers:
>     * *"...which is **technically reasonable and novel** within the scope of TTA."* (by Reviewer w35R)
>     * *"**The methods** themselves are not novel but **their combination and fit to "wild" TTA are sufficiently novel.**"* (by Reviewer Ww1T)

---

> > ### Author Response · Authors · 2022-11-13
> > **Responses to Reviewer 7Sq7 [2/3]**
> >
> >
> >
> > > Q2. The entropy selective term looks a lot like the one from EATA. It is also worrying to me that the authors do not mention nor discuss this similarity. The sharpness-aware minimization technique is also not new. It is adopted directly from SAM from the supervised loss to the self-supervised loss.
> >
> >
> > A2. In this paper, our SAR seeks to stabilize online test-time adaptation (TTA) under wild test settings. Our goal is different from 1) EATA which seeks to improve the efficiency of TTA and 2) SAM which seeks to improve the model generalization ability by training-time optimization. The **novelty of our method is highly recognized by Reviewers AHGC, w35R and Ww1T**: 1) *"Their method is **quite novel**"*，2）*"... is technically **reasonable and novel within the scope of TTA**"* and 3) *"... fit to wild "TTA" are **sufficiently novel**"*. We discuss the detailed relations between SAR and EATA/SAM in the following.
> >
> > 1. Differences from EATA:
> >     * EATA seeks **to improve the adaptation efficiency** via sample entropy selection.
> >     * In our SAR, we discover that some noisy/large gradients may hurt the adaptation and thus result in model collapse under wild test settings. Then, we exploit an **alternative** metric (i.e., entropy selection) **to remove partial noisy/large gradients**. However, this is **still not sufficient for achieving stable TTA** (see ablation results in Table 5). Thus, we further introduce the sharpness-aware optimization and a model recovery scheme. With these three strategies, our SAR performs stably under wild test settings.
> >
> > 2. Differences from SAM:
> >     * SAM (sharpness-aware minimization) aims to improve the model generalization ability at **training time**, by conducting sharpness-aware learning with supervised losses.
> >     * In our SAR, we seek to alleviate the effects of noisy/large gradients that may result in model collapse in online TTA. To this end, we regularize the **test-time** entropy surface to be flat, making the online **model update robust to those remaining noisy/large gradients**. We achieve this by introducing a sharpness-aware learning scheme.
> >
> > We have included these discussions in Section A of the revised paper.
> >
> >
> >
> > > Q3. On a side note, the authors mentioned a problem with gradient selection/clipping is that the threshold needs careful tuning. However, I would also argue that the hyper-parameter for the sharpness-aware entropy minimization also needs careful tuning, since it should depend on the norm of the parameters of the source-pretrained model.
> >
> > A3. Thank you for your valuable suggestion. Actually, the **hyper-parameter $\rho$** in sharpness-aware optimization **is not hard to tune in our online test-time adaptation**. Following Foret et al. (2021), we set $\rho=0.05$ in all experiments, and it works well with different model architectures (ResNet50-BN, ResNet50-GN, VitBase-LN) on different datasets (ImageNet-C, ImageNet-R, VisDA-2021). We also conduct a sensitivity analysis of $\rho$ in Table A, in which our SAR works well under the range [0.03, 0.1].
> >
> > Table A. Sensitivity of $\rho$ in our SAR. We report **accuracy (%)** on ImageNet-C (shot noise, severity level 5) **under online imbalanced label distribution shifts.**
> > | No adapt | Tent | SAR ($\rho$=0.01) | SAR ($\rho$=0.03) | SAR ($\rho$=0.05) | SAR ($\rho$=0.07) | SAR ($\rho$=0.1) |
> > |:--------:|:----:|:-----------------:|:-----------------:|:-----------------:|:-----------------:|:----------------:|
> > |   6.7    | 1.4  |       40.3        |       47.7        |       47.6        |       47.2        |       47.2       |

---

> > > ### Author Response · Authors · 2022-11-13
> > > **Responses to Reviewer 7Sq7 [3/3]**
> > >
> > >
> > > > Q4.The authors only validate their method for the case of entropy minimization. To the best of my knowledge, there are more surrogate losses that are often used for ODA. It would be beneficial to test if the method also helps in these cases.
> > >
> > > A4. Thank you for your valuable question. We would like to answer this question from the following two aspects:
> > >
> > > 1. **Why we do not combine our method with other Online Domain Adaptation losses**. In this paper, we study online test-time adaptation (TTA). In this task, the **conventional online domain adaptation losses are not appropriate** due to the following reasons:
> > >     * In our online TTA, we adapt a given model exploiting only unlabeled test data. Conversely, conventional online domain adaptation may rely on accessible the source training data (such as [1]) or some ground-truth labels of the target data (such as [2]).
> > >     * In our online TTA, we adapt a model instantly for each incoming test sample or a mini-batch. In contrast, conventional online domain adaptation may rely on plenty of target/test samples (multiple batches) at each time step, such as [1].
> > > 2. **Why we do not devise our method based on other Online Test-Time Adaptation losses**. In this paper, we devise our SAR based on **entropy minimization** since it is the **most efficient** in the current online TTA community and also has **excellent performance**. To be specific,
> > >     * Many other online TTA losses rely on multiple forward-and-backward propagations for each test sample and thus are computationally inefficient at test time, such as rotation prediction (in TTT), contrastive loss (in TTT++) and prediction consistency maximization (in MEMO). Moreover, the contrastive loss in TTT++ relies on negative sample pairs, and thus may be hard to handle the case that only one test sample arrives at each time step.
> > >
> > > [1] Online Unsupervised Domain Adaptation via Reducing Inter- and Intra-Domain Discrepancies. IEEE TNNLS, 2022.
> > >
> > > [2] Active Online Learning with Hidden Shifting Domains. AISTATS, 2021.
> > >
> > >
> > >
> > > > Q5. I wonder why the authors do not test their method with BN models (e.g., original resnet50)? If it does not perform well in this case, I suggest adding some explanation/discussion regarding this.
> > >
> > > A5. Thank you for your valuable suggestion. Prior TTA methods often adapt models with batch norm (BN) layers and are often built upon BN statistics adaptation. However, these methods would fail under wild test settings. In this paper, we analyze the unstable reasons for online TTA methods under wild test settings and discover that **BN is a crucial factor hindering TTA's stability**. Therefore, **we devise our SAR based on models with batch-agnostic norm layers (GN and LN) instead of BN**, as described in Section 3.1. We also provide the result of SAR on the BN model in Table B of Q6.
> > >
> > >
> > >
> > > > Q6. It would also be beneficial to conduct experiments on domain shift datasets such as VisDA.
> > >
> > > A6. Thank you for your valuable suggestion. We further compare our SAR with online TTA methods (Tent and EATA) on VisDA-2021. As shown in Table B, results are consistent with that on ImageNet-C and ImageNet-R. Specifically, Tent and EATA fail with the BN model (ResNet50-BN) while performing stably with the GN/LN model (ResNet50-GN/VitBase-LN). Besides, our SAR further improves the adaptation performance of Tent and EATA on ResNet50-GN/VitBase-LN. These results further verify the effectiveness of our SAR.
> > >
> > > Table B. Classification **accuracy (%)** on VisDA-2021 **under online imbalanced label distribution shifts** and **mixed domain shifts**.
> > > | Model       | No adapt | Tent | EATA | SAR (ours) |
> > > | ----------- |:--------:|:----:|:----:|:----------:|
> > > | ResNet50-BN |   36.0   | 9.0  | 7.4  |    9.8     |
> > > | ResNet50-GN |   43.7   | 42.8 | 40.9 |  **44.1**  |
> > > | VitBase-LN  |   44.3   | 49.1 | 49.0 |  **52.0**  |
> > >
> > >
> > > We sincerely hope our clarifications above have addressed your concerns and can improve your opinion of our work.

---

> ### Author Response · Authors · 2022-11-17
> **Thanks to Reviewer 7Sq7**
>
> Dear Reviewer 7Sq7,
>
> We would like to thank you again for your valuable feedback to improve our work. We are wondering whether our response has addressed your questions and can improve your opinion of our work.
>
> Kindly let us know if your might have further comments, and we will do our best to address them.
>
> Best,
>
> The Authors

---

> > ### Author Response · Authors · 2022-11-24
> > **Kind reminder for discussion**
> >
> > Dear Reviewer 7Sq7,
> >
> > We have provided point-by-point responses to your concerns. We believe that our responses have addressed your concerns but still haven’t gotten any feedback from you. Do you have any further comments/suggestions?
> >
> > Best,
> >
> > Paper 593 Authors

---

> ### Author Response · Authors · 2022-12-04
> **Thank you for increasing your score! & Further response**
>
> Thank you for your valuable feedback and for upgrading your score! We would like to answer your further questions below.
>
> >Q1. In the comparison with EATA and SAM, the authors pointed out some differences between their proposed method and these existing works. It is absolutely important to acknowledge and discuss these details in the paper. As far as I am aware, the authors did not include any new discussions regarding this in the revision.
>
> A1. Thank you for the valuable suggestion. We discussed the differences between our SAR and EATA/SAM in the Related Work section (Appendix A). We will sharpen our discussions to make them more thorough in the final paper.
>
> >Q2. The authors mentioned they do not apply their method to other test time adaptation method, since they are computationally expensive. I find this arguement not convincing. If some methods are more expensive but offer better performance, they are still worth some value (some people even prefer these methods more for the performance).
>
> A2. We devise our SAR method based on entropy minimization since it is **not only computationally efficient but also has excellent performance**.
> * **Efficiency:** From Table 1 of the main paper, our SAR (entropy-based) is **31$\times$ and 486$\times$ faster** than other non-entropy-based methods TTT and MEMO, respectively.
> * **Performance:** From Table A, our SAR (entropy-based) achieves the best performance in most cases, e.g., SAR outperforms TTT **17.1%** (from 40.9% to 58.0%) regarding the average accuracy on VitBase with severity level 5.
>
> In this work, we focus on entropy minimization and do not further apply SAR to TTT. The reason is that our **SAR is carefully designed for entropy minimization by investigating and addressing the failure cases of online entropy minimization**, and SAR does not explicitly consider the characteristics of TTT loss. We agree that analyzing and improving the stability of other online TTA methods/losses is also an interesting and important future direction. We leave this to our future work.
>
> Table A. Comparison on ImageNet-C under **online imbalanced label distribution shifts**. We report the **average accuracy (%) over 15 different corruption** types.
> | Model+Method                     | Severity level 5 | Severity level 3 |
> | -------------------------------- |:----------------:| :--------------: |
> | ResNet50-GN                      |       30.6       | 54.1             |
> | +TTT (rotation prediction)       |     **41.2**     | 57.7             |
> | +MEMO (consistency maximization) |       31.3       | 54.6             |
> | +Tent (entropy-based)            |       22.0       | 55.4             |
> | +SAR (entropy-based, ours)       |       37.2       | **59.9**         |
> |                                  |                  |                  |
> | VitBase-LN                       |       29.9       | 53.8             |
> | +TTT (rotation prediction)       |       40.9       | 43.5             |
> | +MEMO (consistency maximization) |       39.1       | 62.1             |
> | +Tent (entropy-based)            |       47.3       | 69.5             |
> | +SAR (entropy-based, ours)       |     **58.0**     | **71.3**         |
>
> We sincerely hope our clarifications above have addressed your concerns. Kindly let us know if you have any further comments, and we are happy to continue the discussion.

---

### Author Response · Authors · 2022-11-18
**General  Response**

Dear ACs and Reviewers,

We sincerely appreciate your time and effort in reviewing our paper and providing constructive feedback. Besides the response to each reviewer,  here we would like to further 1) thank reviewers for their recognition of our work and 2) highlight the major modifications in our revision:

1. **We are glad that the reviewers appreciate and recognize our novelty and contributions.**

    *  The proposed setting of "wild" test-time adaptation (TTA) is **realistic** and **of high practical relevancy**, and it is considerably **more challenging**. [w35R, Ww1T]
    * The detailed **investigations of unstable reasons** of TTA under wild scenarios are **interesting** and **provide new insights** for the TTA community. [w35R, AHGC, Ww1T]
    * The proposed SAR method to fit wild online TTA is **technically reasonable, efficient** and **quite novel**. [w35R, AHGC, Ww1T]
    * The paper **is strong on the empirical side**. Extensive experiments and ablation studies are convincing and thorough. [w35R, AHGC, Ww1T]
    * This paper is well-written and easy to follow. [7Sq7, w35R, AHGC, Ww1T]



2. **We summarize the main modifications in our revised paper (highlighted in blue).**

    * We add more results on ImageNet-R and VisDA-2021 in Appendix E. The new results further verify our claims and the effectiveness of our proposed SAR. [7Sq7, Ww1T]
    * We add more ablation results under the wild setting of batch size 1 and mix domain shifts in Appendix F. The new ablate results are consistent with that under online imbalanced label shifts in Table 5. [Ww1T]
    * We add more results of the model recovery scheme with Tent and EATA in Appendix G, which further verify the necessity of our proposed SAR. [Ww1T]
    * We revise the statements regarding the comparison with Gradient Clipping in Section 5.2 to make them clearer. [AHGC]
    * We add discussions and analysis regarding the hyper-parameter $\rho$ of SAR to Appendix F, showing that $\rho$ is easy to tune and insensitive. [7Sq7]
    * We add more discussions regarding the differences between SAR and EATA in Appendix A. [7Sq7]
    * We revise Section 4 to make it more readable, including the font size in figures, clear figure references and statements to describe norm layer effects in TTA. [w35R, Ww1T]

Best regards,

The Authors

---

### Decision · Program_Chairs · 2023-01-20

**Decision:**

Accept: notable-top-5%

**Justification For Why Not Higher Score:**

N/A

**Justification For Why Not Lower Score:**

There is no reason for lower... the paper received great scores, authors provided a thorough response, and the proposed method is simple and working.


**Metareview: Summary, Strengths And Weaknesses:**

This paper proposes a new learning scheme, namely SAL (Sharpness-Aware & Reliable entropy minimization), to tackle the problem of test time adaptation (TTA) under distribution shift. The problem itself is practically important and the paper presents a sound solution that combines entropy minimization and sharpness aware minimization to mitigate TTA instabilities. The final algorithm is computationally efficient and works on real data. A thorough empirical study (including ablations) with encouraging results (both in the paper and its supplementary appendix) show the effectiveness of the approach both in terms of its computational efficiency and learning performance. The submission received 4 reviews, 3 of which are on clear accept side. One reviewer was initially more pessimistic about the results, specially around its novelty, but after multiple rounds of question/response between authors and the reviewer, the reviewer increased their score. In concordance with the reviewers, I think this submission provides a clear contribution to TTA and I recommend accept.

**Note From Pc:**

if the above contains the word "oral" or "spotlight" please see: "oral" presentation means -> notable-top-5% and "spotlight" means -> notable-top-25%. As stated in our emails, we are disassociating presentation type from AC recommendations

**Summary Of Ac-Reviewer Meeting:**

N/A